# Global dissociation of the posterior amygdala from the rest of the brain during REM sleep

Marta Matei[1,2], Antoine Bergel [1,2], Sophie Pezet [1] & Mickaël Tanter [1✉]

Rapid-eye-movement sleep (REMS) or paradoxical sleep is associated with intense neuronal activity, fluctuations in autonomic control, body paralysis and brain-wide hyperemia. The mechanisms and functions of these energy-demanding patterns remain elusive and a global picture of brain activation during REMS is currently missing. In the present work, we performed functional ultrasound imaging on rats over multiple coronal and sagittal brain sections during hundreds of spontaneous REMS episodes to provide the spatiotemporal dynamics of vascular activity in 259 brain regions spanning more than 2/3 of the total brain volume. We first demonstrate a dissociation between basal/midbrain and cortical structures, the first ones sustaining tonic activation during REMS while the others are activated in phasic bouts. Second, we isolated the vascular compartment in our recordings and identified arteries in the anterior part of the brain as strongly involved in the blood supply during REMS episodes. Finally, we report a peculiar activation pattern in the posterior amygdala, which is strikingly disconnected from the rest of the brain during most REMS episodes. This last finding suggests that the amygdala undergoes specific processing during REMS and may be linked to the regulation of emotions and the creation of dream content during this very state.

[1] Physics for Medicine Paris, Inserm U1273, ESPCI Paris, CNRS UMR 8063, Paris Sciences et Lettres research University, Paris, France. [2] These authors contributed equally: Marta Matei, Antoine Bergel. ✉email: mickael.tanter@espci.fr

Sleep is ubiquitous in animals, yet its functions remain unknown. Interestingly, most species, encompassing insects, fishes, birds and mammals exhibit two types of sleep: non-REM sleep and REM-sleep (REMS)[1]. Though the first one is characterized by quiescent brain activity and low-energy expenditure, the second is actually associated with 'activated' brain state, phasic muscular activation despite body paralysis and increased energy consumption and metabolic load[2]. This last observation strongly challenges the view of sleep as a passive state. Moreover, the discovery of REMS has triggered a paradigm shift in considering sleep as a state during which critical operations are performed despite apparent behavioral quiescence[3,4]. Over the past decade, sleep researchers have accumulated evidence showing that REM sleep is a complex state produced by anatomically distributed neural circuits, serving a wide variety of functions critical for emotional regulation, memory and development[5–7]. However, to build a unified understanding of this complex state, a clear global picture of brain activity during spontaneous REM sleep is critical, but currently missing.

From a physiological point of view, complex brain circuits have been shown to play a role in REMS. For example, in the brainstem[8], two subsets of glutamatergic neurons (so called REM-on neurons) located in the latero-dorsal tegmental area and in the sub-latero-dorsal tegmental nucleus have been described: the first one projecting to the forebrain, responsible for the generation of hippocampal theta rhythm and desynchronized cortical activity, the second one projecting to the brainstem and responsible for muscle atonia hence the suppression of motor activity[2]. Furthermore, the melanin concentrating hormone, known to play a role in the promotion of REMS, is synthesized by neurons in the hypothalamus[9]. However, the downstream activations of these pathways are completely unknown. Taken together, these examples show that REMS is a complex brain state which involves many brain regions scattered across the brain, rendering the global investigation of REMS challenging. Apart from the hippocampal structure[10–12], electrophysiological data during REMS remains sparse. Though some fMRI/PET studies have investigated functional connectivity associated with REMS[13,14], the limited temporal resolution and complexity of these techniques impedes the characterization of whole-brain networks during single REMS episodes.

Several studies have found a link between REMS and emotions, such as emotions processing[5,15,16], emotions recalibration[17,18] and REMS disturbances related to post-traumatic stress disorder emotional dysregulation[19–22].

In addition to the regulation and processing of emotions, several other functions have been attributed to REMS. In particular, hippocampal theta oscillations, that can be disrupted optogenetically by the selective inhibition of GABAergic neurons in the medial septum, are instrumental in the formation of hippocampus-dependent memories in a novel object recognition task and contextual fear-conditioning in mice[6]. Additionally, REM sleep is critical for the brain maturation, specifically during early life REMS[23] and more precisely to aid sensorimotor system development through muscle twitches[24]. Another function attributed to REMS is memories forgetting (ref.[25], see[26] for a review), probably through the activation of melanin-concentrating-hormone-producing neurons[27]. In the same line of thinking, REMS has been shown to be crucial for the selective synaptic pruning of irrelevant synapses and the maintenance of new synapses in the developing mouse cortex (Li et al 2017).

Interestingly, our lab has previously demonstrated, using functional ultrasound (fUS) coupled to local field potential (LFP) recordings in rats, an intense hyperemic activity largely exceeding both NREMS and wake levels[28]. These large-amplitude hyperemic patterns in the hippocampus, thalamus and cortex occurred in phasic bouts, followed a sequential thalamus-hippocampus-temporal cortex pattern and were robustly preceded by fast-gamma oscillations in the CA1 region. Such intense hyperemic activity was also imaged using functional ultrasound in human neonates during active sleep[29]. Another recent study confirmed this hyperemic activity during REMS, using intrinsic optical imaging in sleeping head-fixe mice[30]. Notably, these findings are in line with the entrainment of arteriole diameter by gamma activity in the cortex of head-fixed mice during wake[31]. In both studies however, the characterization of neurovascular interactions was restricted to the cortex. These findings are puzzling because sleep is generally thought to be a period when the body is resting and its energy restored[32–34], and yet this hyperemic activity seems extremely energy consuming. We assume that if such energy-demanding activity has been maintained across evolution, it must have some important role for the survival of the animal, which does not seem to have been found yet.

As previous studies did not provide a global view of REMS brain activity, we took advantage of fUS versatility to sequentially scan >250 brain regions over multiple coronal and parasagittal sections (2/3 of the total rat brain volume, within hundreds of REMS episodes). This study thus not only provides an exhaustive and more complete characterization of global brain hemodynamics during rodent REMS due to increased field of view, but also unveils, to the best of our knowledge, novel findings in deep brain structures. We demonstrate that REMS hyperemia is especially pronounced in the medial and posterior part of the brain. Second, we show a clear dissociation between basal/mid-brain structures and superficial ones, respectively activated in a tonic and phasic manner. Third, we disentangle the vascular structures involved in the irrigation of the brain during REMS episodes providing a detailed outlook of blood supply. Finally, we show that brain activity reveals a striking dissociation between the posterior part of the amygdala complex and the rest of the brain regions. Taken together, these results provide an important resource for REM sleep researchers for the years to come and will enrich current models of REMS function.

## Results

This study aimed at investigating the large-scale hemodynamics during REMS, in particular deep structures that are not easily accessible by state-of-the art techniques. Using an experimental approach developed previously[35], which included a cranial window, the implantation of local field potential (LFP) electrodes and a permanent fUS-compatible plastic prosthetic skull, and the mounting of the ultrasound probe holder (Fig. 1a). In this setup, the different regions of the brain were monitored in a series of coronal and sagittal planes, each acquisition lasting 30 min for 4–6 h per day over the course of several days (Fig. 1a). This resulted in a dataset of 72 recordings in $n = 3$ animals, totalizing 577 REM episodes recorded in 259 brain regions, together with hippocampal LFP recordings, accelerometer, and neck electromyogram (EMG) (see Supplementary Table 1 and Supplementary Figs. 1 and 2 for data acquisition and Supplementary Fig. 3 for electrode implantation and identification).

**Volumetric Reconstruction of REMS vascular networks.** We performed measurements of the cerebral blood volume (CBV) in $n = 3$ rats, in a total of 72 recordings, during which animals are spontaneously going through different vigilance states (quiet wake—QW, active wake–AW, non-REM sleep—NREMS, REM sleep—REMS, see Methods for details regarding the classification). In order to render the mean activations levels for all brain structures and states, we performed a series of processing steps that included: (1) data acquisition along coronal and sagittal brain sections,

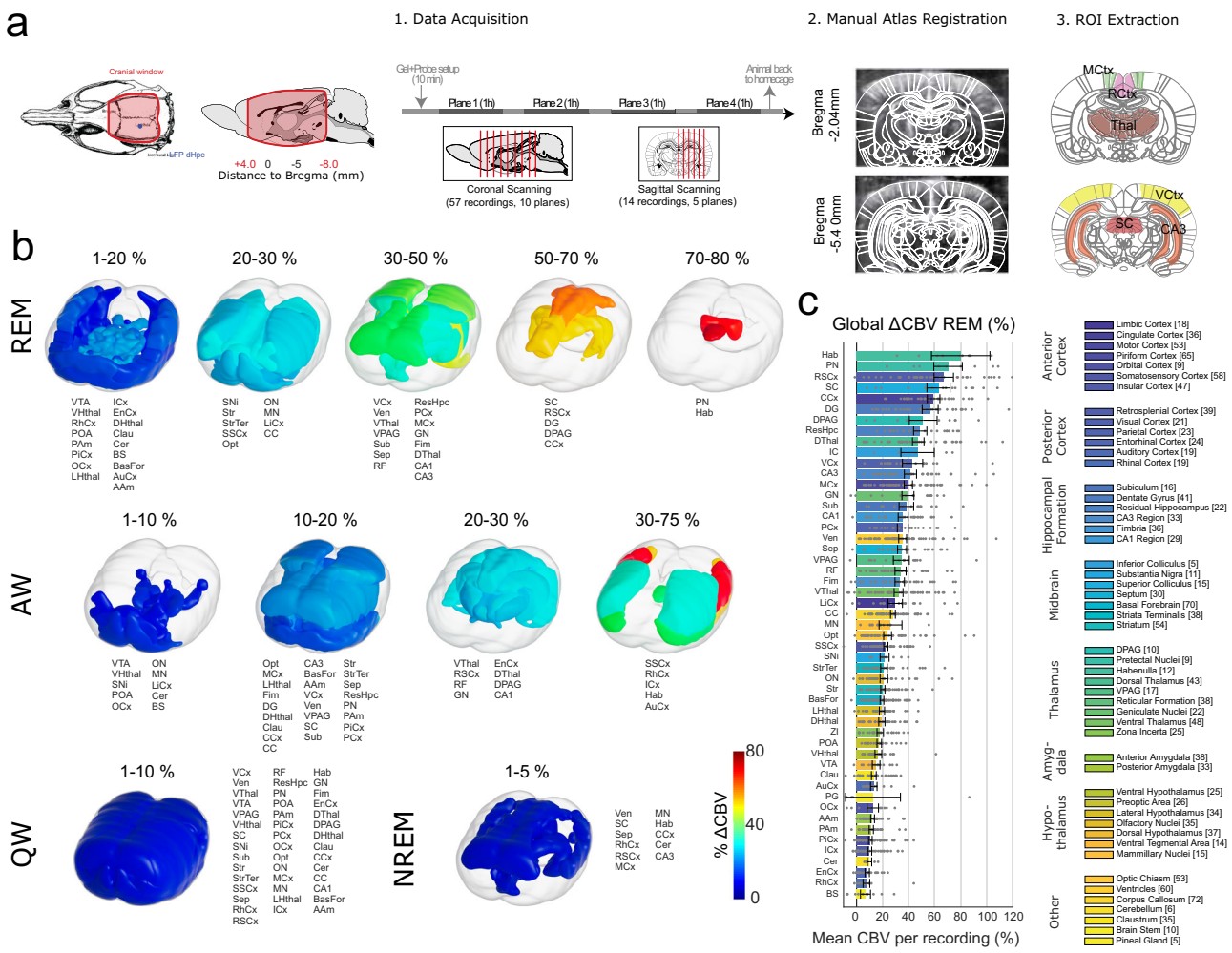

**Fig. 1 Volumetric reconstruction of REM sleep network based on sequential planar functional ultrasound imaging. a** Data acquisition and analysis pipeline (1) Large cranial window and LFP electrode location. Such chronic preparation yields a large field of view and allow for sequential scanning of the brain on coronal (57 recordings, 10 planes, 3 rats) and parasagittal (14 recordings, 5 planes, 3 rats) planes. (2) A reference atlas is manually registered on each recording using salient landmarks. (3) Masks corresponding to regions of interests are then used to compute mean regional cerebral blood volume (CBV) for defined periods of time. Mean CBV values are then computed across all recordings and rendered in a 3D brain volume. **b** Volume rendering of the mean values of CBV changes for 4 vigilance states (QW: quiet wake, AW: active wake, NREMS: non-REM sleep, REMS: REM sleep). Note that CBV levels during REMS largely outmatch all other regions. Only regional values above 1% are shown for clarity. Note the strong activations in the superior colliculus, retrosplenial cortex, dentate gyrus, dorsal part of the PAG, cingulate cortex, pretectal nuclei and habenula. **c** Bar plots showing the mean CBV changes distributions across regions of interest. Regions have been grouped and color-coded according to relevant brain structures. Gray dots are individual recordings, error bars display standard-deviation (72 recordings, 3 rats). Atlas images adapted from The Rat Brain in Stereotaxic Coordinates: Compact, Paxinos G. and Watson C., Copyright (2017), with permission from Elsevier.

(2) manual registration onto an anatomical atlas using salient landmarks, (3) ROI extraction and sleep scoring to finally compute regional averages of CBV time-series for all vigilance states and finally to re-project the mean values onto a 3D volume (Fig. 1a). This revealed a quiescent level of CBV fluctuations in QW and NREMS in all brain structures. AW however is associated with increased cortical CBV levels especially in the primary sensory areas, while REMS is characterized by increased CBV in all brain regions with strongest effect in the hippocampal and limbic structures (Fig. 1b). As we aimed at studying in detail the hemodynamic changes in various parts of the brain during REMS, we next quantified the changes in 56 regions of interest located under our various imaging planes, by computing the percentage of CBV change during REMS (Fig. 1c). Calculations were both performed using 1–3 min of either the QW or the AW for the baseline (Supplementary Fig. 4). This double analysis shows consistently a large range of CBV changes in association with

REMS between different parts of the brain. While the hippocampal formation, the periaqueductal gray (PAG), the superior colliculus (SC) and some parts of the cortex, (such as the cingulate and retrosplenial cortices) present a strong percentage of CBV increase during REMS, areas of the hypothalamus and laterally located cortices (auditory, rhinal, piriform cortices) present modest CBV increases during REMS. This combined analysis demonstrates that REMS hyperemia is not only a state of intense activation with respect to QW and NREMS, which are known to quiescent states, but also to AW in all brain regions, with strongest effects in the hippocampus and midbrain structures. Detailed mean values of the CBV distributions in all regions across the different vigilance states are detailed in Supplementary Data 1.

**Spatial and temporal heterogeneities of REMS activations throughout the brain.** We have previously described strong

hemodynamic changes associated with REMS over a single coronal plane (Bregma −4.0 mm), which were composed of both phasic and tonic components[28]. Here, we aimed to assess whether such tonic/phasic dissociation was also present in other recording planes. Interestingly, we found a peculiar dissociation between superficial and deep pixels over the same coronal section in the frontal part of the brain (Bregma +3.0 mm, Fig. 2a). By thresholding vascular activity in all brain pixels during REMS (which was set independently for each pixel based on the mean and standard-deviation of its distribution during active wake, see Methods), we were able to classify pixels as 'active' or 'inactive' during a single REMS episode, which allowed us to define time intervals as REM-PHASIC periods, that we also refer to as vascular surges (VS), when >50% of brain pixels were active simultaneously for >3 s (Fig. 2b and Supplementary Fig. 5 for the mean correlation maps for each recording). This enabled us to extract a binary variable that accounted for the phasic (respectively tonic) component of REMS (seed phasic-REM, equals 1 during phasic activity, 0 otherwise). We then used these two variables as 'seeds' for correlation analyses shown later. This process of thresholding individual pixels allowed us to compute instantaneous activation maps (simply a binary image resulting from a thresholded CBV frame) and mean activation maps computed either during REM-PHASIC or REM-TONIC time intervals, which reveals more spread-out and intense activations during REM-PHASIC than REM-TONIC (Fig. 2c). We then investigated spatial recruitment as the proportion of active pixels in a given ROI for a specific time period and showed that cortical and amygdalar regions were poorly recruited during REM-phasic, in contrast with hippocampal and thalamic regions recruited both during tonic and phasic periods (Fig. 2d). Ultimately, we subdivided our coronal recordings in 3 major groups: anterior, intermediate and posterior (respectively anterior to Bregma +0.0 mm, between Bregma +0.0 mm and Bregma −3.0 mm, and posterior to Bregma −3.0 mm) as well as 2 groups for the sagittal recordings: medial and lateral (closer or >2.0 mm from the midline). We then computed the mean number of surges per REMS episode, the percent of phasic REMS per episode, and the mean duration and spatial recruitment of each surge over the whole brain (called surge extent) (Fig. 2e). We show that phasic vascular events are as frequent in all groups but longer and implicate more pixels in the medial and posterior parts of the brain.

**Correlation and GLM Analysis shows a topographical dissociation between cortical and subcortical structures.** In order to assess whether activity within individual voxels taken in the superficial structures were associated with a tonic activation (sustained during a single REMS episode) while superficial pixels were active intermittently by phasic bouts, we performed two comparative analyses. We first computed the different cross-correlations functions obtained with seed-phasic REM and seed-tonic REM for all pixels across all planes and extracted maximal correlation maps. This allowed us to compute differential maps between tonic and phasic periods which shows a stronger involvement of superficial regions during phasic REM sleep and deep regions during tonic REM sleep (Fig. 3a). This phenomenon was clearly visible on all 4 correlation maps (each pixel displays the maximum of the cross-correlation function) generated with both seeds: cortical structures were more strongly associated with REMS-phasic than with REMS (black arrows) on all brain sections. This effect was confirmed in regional analysis across individuals and interestingly the timing associated with either seed variable yielded different information (Fig. 3b). In particular, timings associated with seed-REM captured the broad inter-episode fluctuations while those associated with REM phasic,

revealed a precise sequence of activation between brain regions and captured the intra-REM fluctuations (Fig. 3b). Moreover, a few brain regions stand out with very high correlation scores such as the dorsal periaqueductal gray (DPAG) ($R_{max\ REM} = 0.743 +/ −0.023$, $N = 3$ animals), the inferior colliculus (IC) ($R_{max\ REM} = 0.730 +/−0.058$, $N = 3$ animals), the substantia nigra (SNi) ($R_{max\ REM} = 0.714 +/−0.015$, $N = 3$ animals), the ventral hypothalamus (VHThal) ($R_{max\ REM} = 0.707 +/−0.016$, $N = 3$ animals), the superior colliculus (SC) ($R_{max\ REM} = 0.697 +/ −0.036$, $N = 3$ animals) as was seen in Fig. 3b. In the correlation analysis however, all cortical regions, except limbic ($R_{max\ REM} = 0.583 +/−0.048$, $N = 3$ animals), cingulate ($R_{max\ REM} = 0.578 +/ −0.020$, $N = 3$ animals) and retrosplenial ($R_{max\ REM} = 0.583 +/ −0.028$, $N = 3$ animals) cortices displayed lower correlation coefficients than other brain structures (Supplementary Data 2). In order to cross-validate our approach, we also performed a simple generalized linear model (GLM) analysis to explain the variance observed during REMS periods with the two orthogonal regressors defined previously (seed REM-phasic, seed REM-tonic) which guarantees the stability of such analysis. This also confirmed that CBV activations could be linearly decomposed into a tonic subcortical component and phasic cortical one as both individual pixels and regional analysis showed a stronger value for the cortical regressors in the phasic map, in comparison with other subcortical structures. In Fig. 3c, we display ratio maps (displayed in log scale) between the two regressors (REM-phasic/ REM-tonic) for all pixels in the four same recording sessions as in (A). We observe up to 2 orders of magnitude difference between superficial and deep pixels on most planes. Finally, the same ratio is computed for large brain areas, showing once again than phasic REM-sleep is best accounted for in superficial structures rather than in deep structures, in particular the hypothalamus and amygdala (Fig. 3d). This last argument establishes REMS as a state of sustained subcortical activation, interleaved with strong phasic events spreading to superficial cortical structures.

**Selective contribution of vascular dynamics to the blood supply during REM sleep.** The second major aim of this study was to elucidate the contribution of the vascular compartment. For this analysis we added 5 rats and 12 recordings to the initial 3 rats and 72 recordings to increase the number of recordings targeting planes with vessels of interest. We first segmented all salient vascular structures in our imaging planes including parallel branches of the main cerebral arteries (anterior cerebral artery—acer, anterior choroidal artery—ach, middle cerebral artery—mcer, and posterior cerebral artery—pcer) and segments along on the anterior branch (anterior cerebral artery—acer, azygos of anterior cerebral artery—azac, azygos pericallosal artery—azp) (Fig. 4a, b). We then investigated the temporal dynamics in these structures by re-aligning their time course to the start of each REMS episode (defined by hippocampal theta activity crossing a threshold) and averaged their activation profile, both from the onset and of REMS (Fig. 4c–f). Quantifications show an increased CBV (expressed in percentage of variation relative to the QW baseline) at the beginning of REMS, a sustained level throughout the REM episode and finally a sudden drop at the end of the REM episode. This increased CBV in arteries was more pronounced (2-fold increase) in the arteries that vascularize the rostral part of the brain (acer, azac and azp), compared to the arteries that vascularize the medial and posterior parts of the brain (ach, mcer, pcer), confirming a general phenomenon of increased blood supply during REMS, but also an emphasis of this enhanced blood flow in the rostral part of the brain. Further analysis shows an important propagation delay along the anterior branch with acer peaking earlier than azac and azp (acer: $t1 = −3.14 +/−$

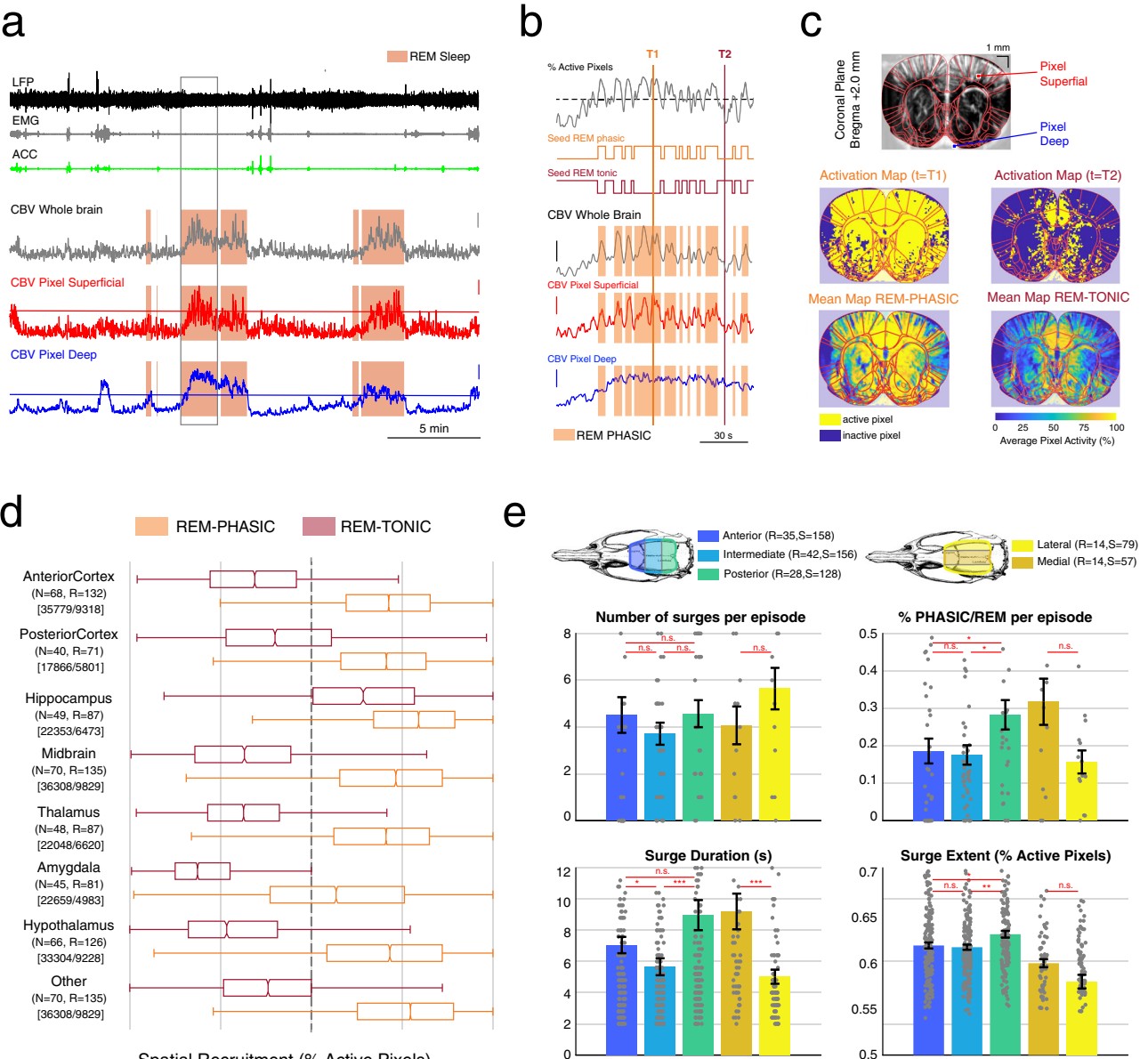

**Fig. 2 Spatial distribution and heterogeneities of phasic vascular events across the brain. a** (Top to Bottom) Typical time-series for a 30-min recording including raw hippocampal LFP (black), neck EMG (gray), accelerometer (green) and cerebral blood volume (CBV) averaged over the whole brain (gray), in a superficial pixel (red) and in a deep pixel (blue). For these two latter time-series, the activation threshold used to detect vascular surges is displayed horizontally (see Methods). Note how superficial pixel activity oscillates around the activation threshold whereas the deep pixel's stays well above threshold. Vertical bar 20% ΔCBV. Horizontal bar 5 min. **b** Details of a single REMS episode, box shown in **a**. (Top) Time-series of the proportion of active pixels within all brain pixels in the recording plane with 50% threshold (dashed line). REM-phasic epochs are defined as time intervals with >50% of active pixels (orange) whereas REM-tonic epochs are time-intervals with <50% of active pixels (brown). (Bottom) Details of CBV in the whole brain, superficial and deep pixels shown in **a** overlaid with REM-phasic epochs (dark red boxes). **c** (Top) fUS recording plane (Bregma +2.0 mm) with Paxinos atlas manually registered and overlaid in red as well as the locations of superficial and deep pixels, which activity is shown in (A-B) (Bottom) Instantaneous and mean activation maps computed for REM-phasic (top) and REM-tonic (epochs). Instantaneous activation maps display the proportion of active pixels (yellow) within a single temporal frame (left) and mean activation maps display the above-threshold activity during a single recording. **d** Box plots showing the distribution of spatial recruitment across 8 major brain regions during REM-phasic and REM-tonic epochs, for all REMS episodes. Note the strong difference between tonic and phasic distributions in the cortical structures, as opposed to non-cortical ones. Box plots show respectively the minimal (left whisker), 25% quartile (left notch), median (middle bar), 75% quartile (right notch) and maximal value (right whisker). **e** Heterogeneities in vascular surge distributions, durations and extent according to their location in the brain. Coronal recordings were subdivided in 3 groups (anterior, intermediate and posterior) and sagittal recordings in two groups (medial and lateral). REM-phasic events occur as frequently in all five groups but there was a marked increase in the percentage of phasic REM-sleep for posterior and medial groups, as well as longer and more spatial extended vascular events according to other groups. Gray dots show individual values for each REMS episode (top) or vascular surge (bottom). Error bars represent mean + / − standard-error of the mean. Statistical testing was conducted under the null hypothesis 'no difference between the mean values observed in the two groups' via Wilcoxon signed rank test (n.s. non-significant: $P > 0.05$, *$P < 0.05$, **$P < 0.01$, ***$P < 0.001$). Exact values are provided as a separate file. Scale bar from **c** = 1 mm. Atlas images adapted from The Rat Brain in Stereotaxic Coordinates: Compact, Paxinos G. and Watson C., Copyright (2017), with permission from Elsevier.

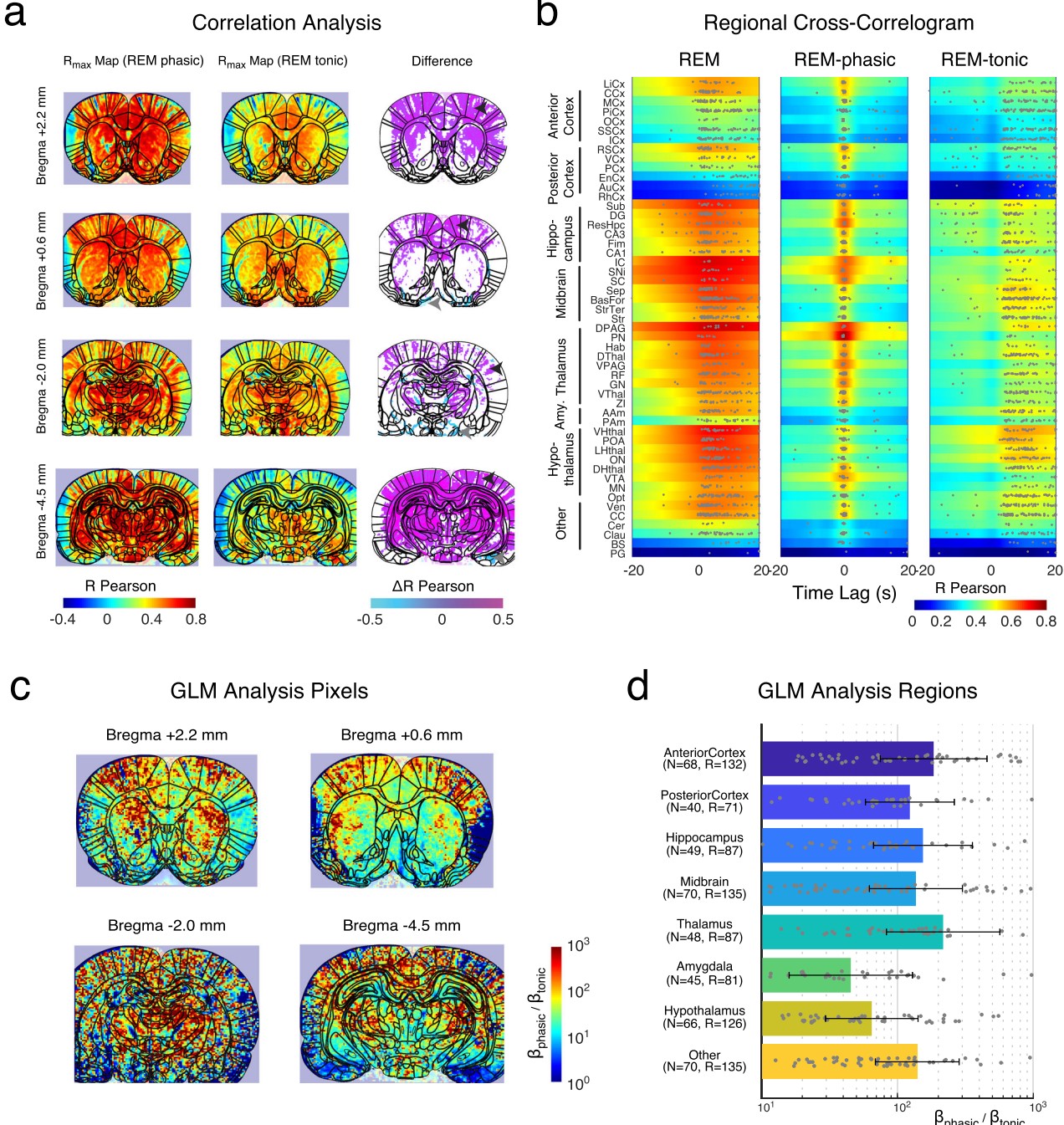

3.26 s, azac: t1 = 1.73 +/− 6.50 s, azp: t1 = 1.81 +/− 3.03 s) (Fig. 4g) Such REM-associated increased CBV was observed at a lower level in veins with a surprising antagonist activity between two side-by-side veins: the longitudinal hippocampal vein (lhiv) and the azygos internal cerebral vein (azicv) (Supplementary Fig. 6).

**Atypical amygdala activity during REM sleep**. When assessing inter-regional correlations in the CBV signal, the most striking pattern of activity was found in the amygdala and consisted of a robust disconnection from the rest of the brain, which was clearly visible on functional connectivity matrices averaged over all REMS episodes (Fig. 5a, see Supplementary Fig. 7 for detailed matrix and Supplementary Fig. 8 for the substructures of the amygdala) and in the temporal fluctuations of individual

recordings (Fig. 5b and Supplementary Movie 1 for a dynamic representation). This effect is consistent with observation from previous figures: the amygdala showing both a relatively low-level of hyperemia during REMS compared to other regions (Fig. 1c) and low-correlation scores (Fig. 3b). Strikingly, the amygdala's activity during REMS, showed a remarkably unique activation profile compared to the rest of the brain which was even greater when comparing the amygdala's anterior and posterior parts, the latter being even more dissociated from the rest of the brain (Fig. 5a). This dissociation could be observed during long periods of strong fluctuation when the remainder of brain activity was silent (Fig. 5b, second part of the episode and Supplementary Movie 1 for a dynamic representation). This effect was confirmed and strengthened using a seed-based approach taking either the regional whole-brain activity as a reference (Fig. 5c) or the amygdala (Fig. 5d), which revealed a very strong and robust

**Fig. 3 Vascular activity during REMS can be decomposed into a tonic subcortical component and a phasic cortical one. a** Maximal Correlation ($R_{max}$) maps obtained by performing correlation analysis between the two seed variables REM-tonic, REM-phasic and all CBV pixel traces in 4 different recording planes in one representative example (at Bregma +2.0 mm, +0.6 mm, −2.0 mm and −4.5 mm). Right Column shows the difference of the two maps (REM-phasic – REM-tonic). Note how superficial pixels display stronger correlation in the REMS-phasic map than REMS-tonic maps, suggesting that they track phasic activations better than deep voxels (black arrows) while deep pixels show the inverse effect (gray arrows). **b** Mean cross-correlation functions computed for the whole set of acquisitions, sorted in decreasing correlation strength by region, for the REM seed (left) REMS-phasic seed (middle) and REMS-tonic seed (right). Hippocampal, midbrain and hypothalamus structures display the strongest coefficient for both analyses with the highest scores for the periaqueductal gray, the substantia nigra and the superior colliculus. Gray dots mark the peak correlation time computed for each recording in each region (the total number of available recordings thus differs from one region to the next). Note the heterogeneity of peak correlation times across regions on the left (and right) correlogram showing that regions are not active simultaneously across a REMS episode, in contrast to the homogeneity in the REMS-phasic (central) correlogram centered on zero-lag, showing stronger temporal synchronization. **c** Regression Coefficient Maps obtained after Generalized Liner Model Analysis performed on REMS frames using the two seed variables REM-tonic, REM-phasic as orthogonal regressors. All maps display the ratio between the two regressors (REM-phasic/REM-tonic) in log scale, as phasic-REM sleep regressors were substantially larger than tonic-REM. Again, note the stronger coefficients in the cortical areas for the phasic regressors (hot colors: strong positive values) in contrast with the deep regions. **d** Regional regressors obtained after GLM analysis on regional time-series variables. Note the strong difference between the amygdala and hypothalamus from the rest of brain structures, in particular anterior cortex and thalamus. Gray dots show individual values for each REMS episode. Error bars represent mean +/ − standard-error of the mean. Atlas images adapted from The Rat Brain in Stereotaxic Coordinates: Compact, Paxinos G. and Watson C., Copyright (2017), with permission from Elsevier.

dissociation between the amygdala and all other brain regions. Interestingly, amygdala sub-structures seem to exhibit also very specific dynamics as shown by the heterogeneous correlation maps found by taking 5 different sub-regions (amygdalohippocampal area, posterolateral part: AHiPL, amygdalohippocampal area, posteromedial part: AHiPm, amygdalopiriform transition area: APir, basolateral amygdaloid nucleus, posterior part: BLP, posterolateral cortical amygdaloid nucleus: PLCo). These results suggest a strong dissociation between the amygdala and all other brain regions during specific epochs of REMS, but also among the amygdala itself.

## Discussion

This study provides a whole-brain characterization of the cerebral and vascular structures involved in the atypical and large-amplitude vascular surges occurring during REMS. This study goes considerably deeper in the understanding of REMS-associated hyperemia, as it imaged a very large number of brain regions (257 regions) over hundreds of REMS episodes.

We implemented fUS imaging in 2D imaging planes with light ultrasonic probes as it is compatible with both unrestrained movement and naturally induced sleep studies. 2D fUS imaging enables us to ensure that the animal is not restrained, behaves almost perfectly normally, and sleeps spontaneously. It is primordial as stressed and head-restrained animals are less eager to sleep; even humans can't go into REM sleep while head-restrained for fMRI studies[36]. Moreover, deprivation protocols are often used to acquire sleep data, which affects both the structure and nature of sleep episodes. As each imaging session could only image on one single 2D plane, we had to repeat the experiment a large number of times in order to achieve an almost full 3D coverage of the brain's regional activity during REMS. Although this approach of multiple 2D planes has the disadvantage to lose the temporal information regarding the coactivity of brain regions from different planes, we solved this difficulty by imaging from both coronal and sagittal planes, thus relying on a respectable number of co-activated regions in each single session.

Although recent technological demonstrations of full 3D fUS imaging using piezo-electric matrix arrays[37,38] or Raw-Colum arrays[39,40] are very promising, they remain to date limited in use, as the heavy weight and limited sensitivity of these probes requires the animal to be head-fixed, thus rendering sleep studies unsuitable and further from normal behavior.

A previous study by our lab has unraveled an intense hyperemic activity during REMS, which largely exceeded both NREMS and wake levels[28]. This hyperemic state is decomposed in a tonic component (the elevation of the baseline) and in a phasic one which is robustly preceded by fast gamma oscillations in the CA1 region. This finding is striking as sleep is supposedly a state in which energy levels are reconstructed[32–34], yet this activity must be highly energy consuming. This hyperemic activity has been confirmed by more recent studies, using intrinsic optical imaging in sleeping head-fixe mice[30]. However, most imaging modalities used so far for REMS studies are either focused on the cortical part or have such a limited temporal resolution that it impedes deeper fundamental understanding.

Such hyperemic activity might be physiologically important as it was kept throughout evolution, despite its energy consumption. Moreover, a clear picture of global brain activity during REMS is still currently missing.

In this study, we used functional ultrasound imaging to gather data on >250 brain regions in both coronal and sagittal planes, thus providing a very exhaustive characterization of global brain hemodynamics during rodent REMS. We demonstrate a clear dissociation between basal/midbrain structures and superficial ones, respectively activated in a tonic and phasic manner. We also disentangle the vascular structures involved in the irrigation of the brain during REMS episodes providing a detailed outlook of blood supply. Finally, one of the most noteworthy results of this work is the striking global dissociation of the amygdala activity from the rest of the brain during the REM episodes.

A previous study has already shown a hyperemic activity during REMS in humans in some brain regions in human using positron emission tomography[41]. However, this study only presented a higher vascular activity correlated with REMS in pontine tegmentum, left thalamus, both amygdaloid complexes, anterior cingulate cortex and right parietal operculum, and some regions with a negative correlation with REMS mainly in cortical areas.

One of the key findings of the present work is that hyperemia is global and spans throughout all of the forebrain that we were able to image (2/3 of total brain volume). Additionally, it was more sustained in the deep/midbrain structures (in particular in the hippocampus) than in the cortex, which activated in phasic bouts. Thus, REMS can be described as a state of tonic hyperemia in the forebrain that only partially spreads to the cortex. Also, activity in the different cortices were strongly heterogeneous, with strongest activations in the retrosplenial, limbic, motor and visual cortices but close to the levels of wake in the other sensory cortices

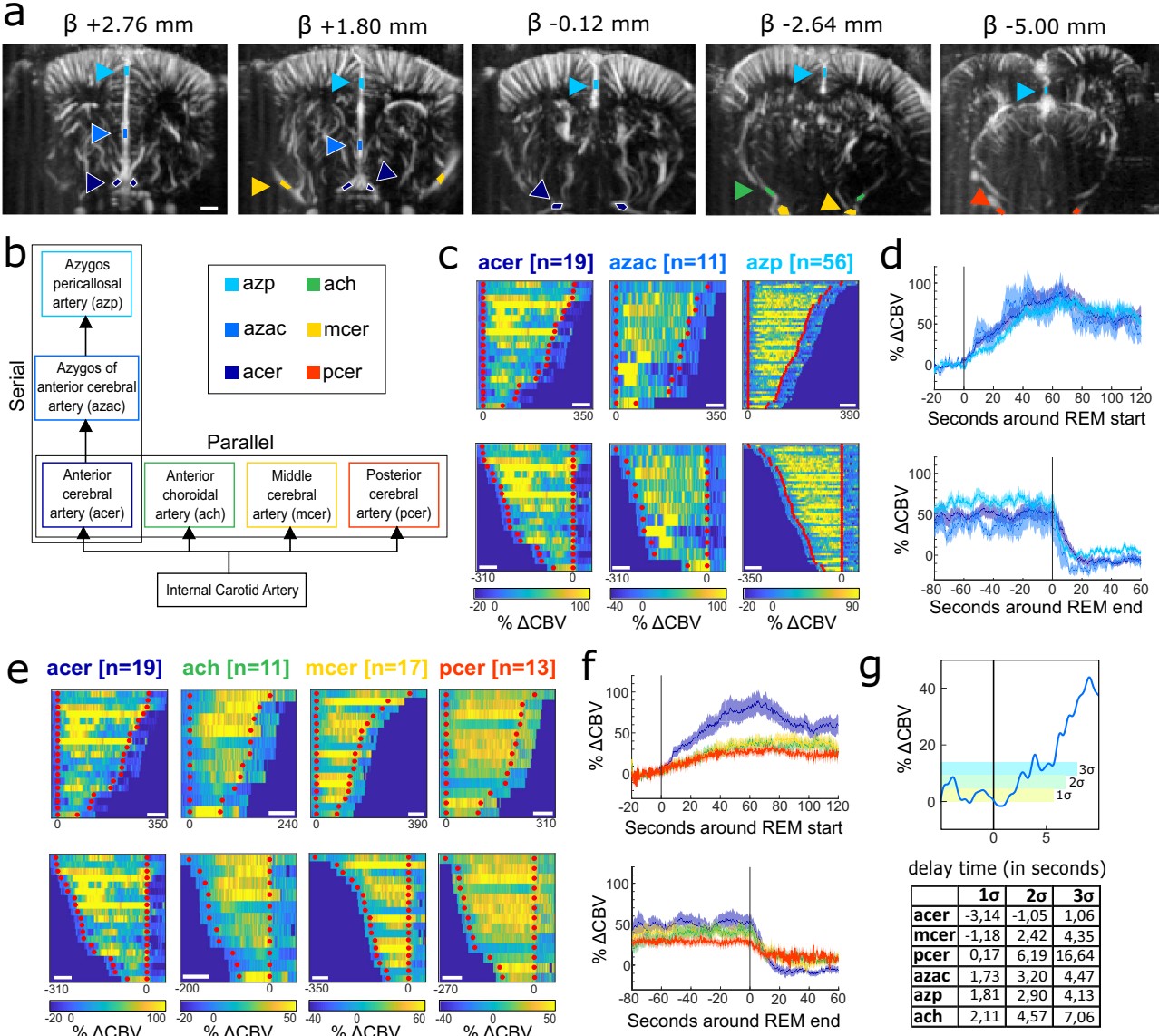

**Fig. 4 Spatiotemporal dynamics of the arterial supply during REM sleep suggests a preferential irrigation in anterior cerebral areas. a** Typical Doppler image (in 5 imaging planes) illustrating the nature and location of the 6 arteries, where the hemodynamic changes were measured, respective to REM episodes (colored arrows, color code: in legend **b**). **b** Hierarchical representation of the 6 different vessels studied, separated in two groups: one within the same branch and the others at the same level and in different branches (according to Xiong et al. 2017). **c** and **e** Representation of the delta CBV in each type of blood vessel over time during all the REM episodes, either aligned to the beginning of REM (top) or the end of REM episodes (bottom). **d** and **f**: averaged delta CBV over time calculated from **c** and **e**, aligned either to the beginning of REM (top) or the end of REM episodes (bottom). Solid line represents the mean and shading represents the distribution. Beginning and end of REM episodes are indicated by red dots in **c** & **e** or vertical lines in **d** and **f**. **g** (top) Averaged representation for the azygos anterior cerebral artery (azac) centered on the beginning of REM and the 3 standard deviations of the averaged signal (σ). The table below shows the averaged times (in seconds) for each vessel to reach the 3 different standard deviations. Scale bar from **a** = 1 mm. Scale bars in **c** and **e** = 60 s. In **c** and **e**: each color bar is applied for the blood vessels, irrespective of their alignment to the beginning or end of the REM episode.

(somatosensory, piriform). This is surprising as rats preferentially use odor and texture rather than vision. Hence, it is possible that hyperemia is associated with the reactivation of visual networks (geniculate, colliculus, cortex) or in link with memory (retrosplenial, septum, and hippocampus). Vascular hyperactivity specific to REMS in rats divides into tonic and phasic regimes, the latter exhibiting transient brain-wide hyperemic patterns, which we called vascular surges (VS). Bergel et al showed that these VS outmatched wake levels occasionally reaching up to a 100% increase in the cortical and hippocampal regions compared to a quiet wake state. Precursors to VS in the theta (6–10 Hz) and

high-gamma (70–110 Hz) bands of hippocampal LFP, and the intensity of each individual VS was best accounted for by the power of fast gamma, suggesting a strong association between local electrographic events and massive brain-wide vascular patterns. These VS exhibit a strong link with LFP gamma power in some brain structures[28]. Although a high correlation was already found in former fUS imaging studies between the fUS signal and EEG recordings[35,42,43] and neural calcium activity[44] in accordance with the neurovascular coupling model[45], the massiveness of the hyperemia during REMS cannot be unambiguously linked to the sole neural activity. Such very high hyperemia may also

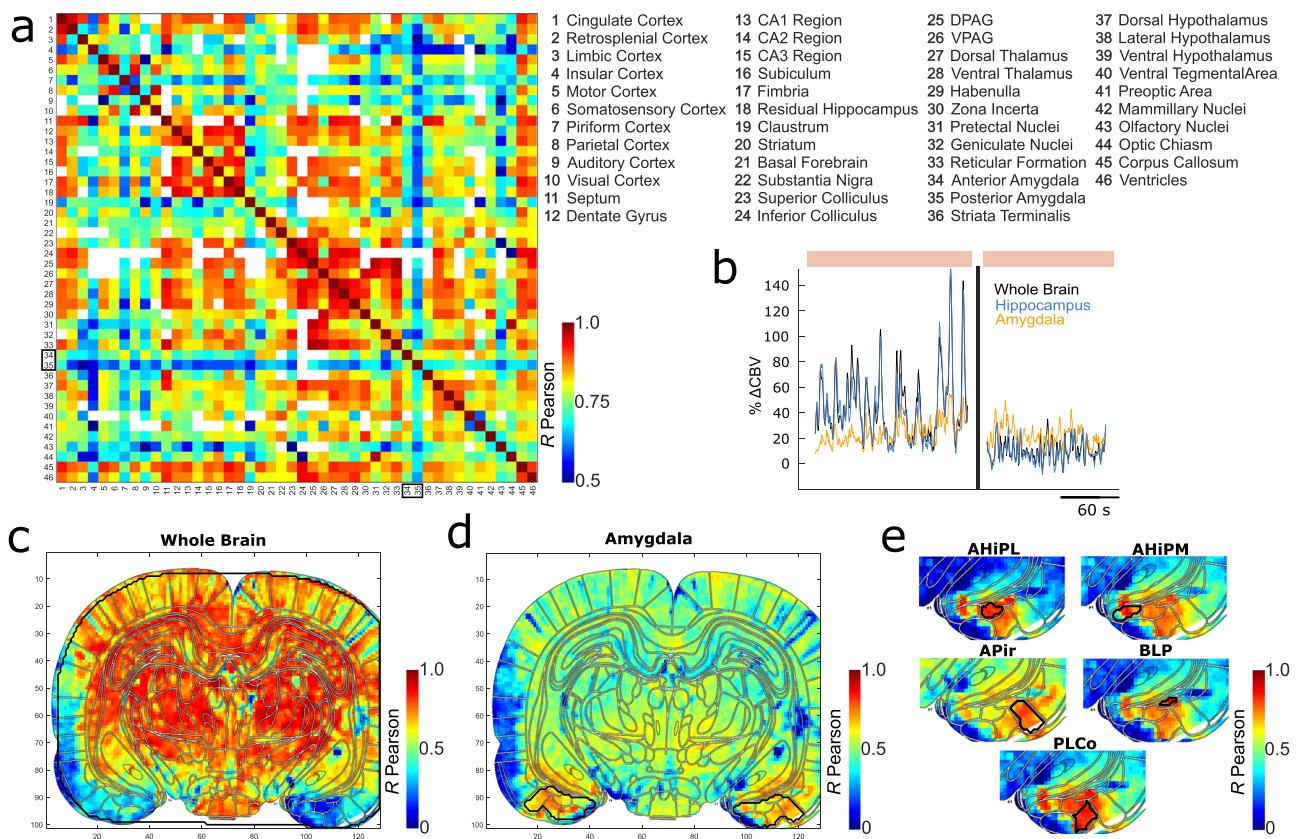

**Fig. 5 Disconnection of the amygdala and sub-structures from the rest of the brain during REMS. a** Connectivity matrix showing in color the average cross-correlation between the 46 different brain regions and nuclei (legend on the right) across all recordings and in all animals ($N = 3$ animals, 72 recordings) (see Supplementary Fig. 7 to have the number of recordings per pixels). White pixels correspond to cross-correlation couples where the number of recordings was either non-existent (because the two brain regions were not present on the same plane) or inferior to 5. Boxes highlight anterior and posterior amygdala. **b**–**e** Example of one specific recording. **b** CBV dynamics of the entire brain (black), the amygdala (yellow) and the hippocampus (blue). The colored patches on the background represent REMS. The dynamics are represented as an averaged value of the percentage of CBV compared to the baseline for each region. Note how the CBV changes dynamics between the first part of this REM episode and the second one: the first part has both a tonic and phasic component while the second part presents only a tonic component. The amygdala keeps the same tonic activity from the first part to the second while the rest of the brain (black) has a slightly lower CBV in the second part. **c**, **d** Pixel correlation maps with respectively, the 'Whole Brain' (**d**) and the 'Amygdala' (**d**) as seeds (highlighted in black). Note how the amygdala is not correlated with the rest of the brain in **c** and correlated only with itself in **d**. **e** Correlation map details for 5 different nuclei of the amygdala as seeds. For each correlation map, the highest possible correlation per pixel is represented ($R_{max}$). Atlas images adapted from The Rat Brain in Stereotaxic Coordinates: Compact, Paxinos G. and Watson C., Copyright (2017), with permission from Elsevier.

partly be linked to the metabolic demand of other cell type cells such as glial cells or the drainage of the glymphatic system occurring during sleep[46].

Though it is known that dreams occur during both NREMS and REMS[47], REMS ones are more vivid and intense, thus raising the assumed link between emotions and REMS. In "The Interpretation of Dreams" by Sigmund Freud, Freud focuses on the importance of dreams and their understanding, as he considered them to be the gate to one's unconscious and emotional state[48]. More recent studies have confirmed this link between REMS and emotions. REMS is necessary for emotions processing[5,15,16] and mostly to recalibrate the emotional response towards negative emotions[17,18]. Moreover, several studies have shown that the memory is better reinforced when associated with an emotionally loaded-memory than with neutral ones[49–51].

It is also now well established that REMS disturbances are often observed in cases of post-traumatic stress disorder (PTSD). Mellman and colleagues observed shorter and more frequent episodes of REMS in trauma-exposed patients compared to non-trauma patients[19]. These findings were later confirmed, along with other observations indicating that REMS disturbances lead to impaired fear extinction learning, which may explain the development and maintenance of PTSD symptoms[20–22].

A theory emerged a few years ago, called sleep to forget, sleep to remember, assuming that memories are constituted of two components: the factual part on one side, which represents the real event as it happened, and the emotional part on the other, which represents the feelings associated with this event as well as their intensity[52]. The factual part is consolidated during REMS, meaning that it will be encoded in the hippocampus (sleep to remember), while the emotional part is depotentialized, which means that the intensity related to the emotion will be reduced (sleep to forget). This is how we can recall negative memories, without reliving the feelings associated with them with the same strength. This phenomenon is considered to be an overnight therapy, necessary to cope with memories related to distressing events, and it would be this process which is disrupted in cases of emotion-based disorders, such as depression or PTSD, thus recalling the link between REMS disorders and PTSD symptoms. Some studies go against this theory, such as Wiesner et al. which shows that the emotional intensity, linked to images with negative emotional content, did not decrease after a night's sleep[53].

The biological mechanisms of the emotional regulation of memories involve an amygdala–hippocampus–medial prefrontal cortex (mPFC) network, whose intercommunication is enhanced by theta and ponto-geniculo-occipital (PGO) oscillations, as well as high levels of acetylcholine and cortisol during REMS[5,54]. These PGO waves have been linked to memory consolidation and also enhanced synaptic plasticity in the amygdala and dorsal hippocampus[55]. Another study also correlated PGO wave density with effective fear extinction learning following trauma[56]. Rats also appear capable of producing PGO waves after direct electrical stimulation of the amygdala[57]. Thus, after several sleep sessions containing REMS, when an aversive memory is triggered during wake the hippocampus will send a signal to the mPFC to inhibit the amygdala. This three-way communication supposedly reduces the emotional response to the memory[52].

Considering the major link between REMS and the regulation of emotions, it is of particular interest to highlight here a strongly dissociated activity of the amygdala compared to the rest of the brain. Indeed, a higher vascular activity in amygdala was already found correlated with REMS using positron emission tomography in humans[41]. Interestingly, although a relatively low hyperemia in the amygdala was found during REM sleep using fUS imaging, our results show that the amygdala vascular activity presents a lower correlation with the rest of the brain regions, depicting a global dissociation of the amygdala from other brain regions.

Moreover, it is known for many years that the amygdala is electrophysiologically active during REMS[58] and was also confirmed more recently during NREMS and more precisely during the reactivation of emotional memories associated to hippocampal sharp wave ripples[59].

A first hypothesis would be that during REMS, the amygdala is activated only when the rest of the brain is not and especially the mPFC, which explains the weak correlation between the amygdaloid complexes and the rest of the brain observed in our connectivity matrix. Rats may need to regularly go through phases of emotional regulation during REMS to cope with the daily accumulation of strong emotional memories. One way to test this hypothesis would be to make recordings of the electrical activity in the amygdala, mPFC and hippocampus simultaneously and to record the PGO oscillations to see if it is possible to correlate the simultaneous activities of these regions with an increase in PGO activity. These recordings should also be made during the next wake to verify that the amygdala are in fact deactivated and to see if there is a correlation between amygdala deactivation and an activation of the mPFC/hippocampus[52,54].

Another hypothesis would be that this decorrelation of the amygdala, coupled with the low amplitude of their vascular activity, could be linked to a defensive mechanism preventing the production of negative emotional content during REMS and preventing its premature termination. Indeed, studies have shown that the generation of consciousness is associated with increased activity throughout the brain[60]. Rapid-eye-movements are associated with dreams, which are themselves considered a form of consciousness. In addition, fMRI studies have shown that the presence of rapid-eye-movement was strongly correlated with the activation of many brain regions such as the oculomotor circuit, the cortico-thalamic sensory system, the language system or even the cholinergic system[36]. Finally, generation of consciousness might require the activation of many brain regions simultaneously[61] as well as for the treatment of emotional responses, which should be available for the whole-brain to process such as in the higher-order theory of emotional consciousness[62] of in the theory of constructed emotion[63]. Taken together, these different studies suggest that the simultaneous activation of many brain regions during rapid-eye-movements in REMS contributes to the generation of a state of consciousness,

necessary for the creation of dreams. Thus, a dissociation of the activity of the amygdala during REMS could prevent their contribution to the creation of negative content (such as fear) in dreams and therefore a potential awaking. This defense mechanism would make it possible specially to preserve the function of the physiological phenomena occurring during REMS, which seem to be energy-intensive (due to the strong increase in blood flow) and which would need to be restarted in the event of premature awaking caused by a nightmare. It is potentially this mechanism that would be defective in cases of development of PTSD, since the latter is associated with REMS disorders.

Finally, the study of REMs (Rapid-Eye-Movements) has been linked to dreams content[60] and the eye movements during dreaming are shown similar to those during wake. In the present study, such analysis is not possible as the camera was set up to record the whole field of view and not specifically the eye movements. However we can hypothesize that the vascular surges described in our previous study[28] are linked to other components of phasic REM sleep, such as muscle twitches, REMs, whisking and penile erections. Indeed, these vascular surges are time-locked to hippocampal theta and gamma bursts during phasic REM sleep (they trail by 1 to 2 s depending on the region)[28] which in turn have been shown to precede fluctuations in arterial pressure[64]. This hypothesis is consistent with an earlier study showing widespread activations (measured by event-related fMRI) are time-locked to REMs in sleeping human subjects[36]. Nonetheless, this link between vascular surges and REMs is yet to be demonstrated through simultaneous fUS recordings and eye movements study (through EOG, regular or infrared cameras).

## Methods

**Animal Surgery**. All animals received humane care in compliance with the European Communities Council Directive of 2010 (2010/63/EU). The experimental protocol used in this study was extensively reviewed and approved by the French CEEA (Comité Ethique pour l'Expérimentation Animale) n°59 Paris Centre et Sud under the reference number 2018061320381023. Male adult Sprague Dawley rats aged 12–13 weeks were first put through a week of habituation with the experimenter and then underwent surgical craniotomy and implant of an ultrasound-clear prosthesis. Anesthesia was induced with 2% isoflurane and maintained with ketamine/xylazine (80/10 mg/kg), while body temperature was maintained at 37.0 °C with a heating pad (Phymep, Paris, France). A sagittal skin incision was performed across the posterior part of the head to expose the skull. We excised the parietal and frontal flaps by drilling and gently moving the bone away from the dura mater. The opening exposed the brain from Bregma +4.0 to Bregma −9.0 mm, with a maximal width of 14 mm. An electrode was implanted stereotaxically and anchored on the edge of the flap. A prosthetic skull was sealed in place with acrylic resin (GC Unifast TRAD), and the residual space was filled with saline. We chose a prosthesis approach that offers a larger field of view and prolonged imaging condition over 4–6 weeks compared to the thinned bone approach. The prosthetic skull is composed of polymethylpentene (Goodfellow, Huntington UK, goodfellow.com), a standard biopolymer used for implants. This material has tissue-like acoustic impedance that allows undistorted propagation of ultrasound waves at the acoustic gel-prosthesis and prosthesis-saline interfaces. The prosthesis was cut out of a film of 250 µm thickness and permanently sealed to the skull. Particular care was taken not to tear the dura to prevent cerebral damage. The surgical procedure, including electrode implantation, typically took 6 h. Animals were subcutaneously injected with anti-inflammatory drug (Metacam, 0.2 mg/kg) and prophylactic antibiotics (Borgal, 16 mg/kg), and postoperative care was performed for 7 days. Animals recovered quickly and were used for data acquisition after a conservative one-week resting period.

**Electrode design and implantation**. Electrodes are based on linear polytrodes grouped in bundles of insulated tungsten wires. The difference with a standard design is a 90°-angle elbow that is formed prior to insertion in the brain. This shape enabled anchoring of the electrode on the skull posterior to the flap. The electrode was implanted with stereotaxic positioning micromotor. The prosthesis was then applied to seal the skull. Four epidural screws placed above the cerebellum and above the olfactory bulbs were used as references and grounds. The intra-hippocampal handmade electrode was composed of 25 to 50 µm diameter insulated tungsten wires soldered to miniature connectors (Omnetics Inc, Minneapolis, US). Eight conductive ends were spaced 0.5 to 1 mm apart and glued to form a 5.5 mm-long, 100–150-µm-diameter bundle. The bundles were lowered in the dorsal

hippocampus (left or right) at stereotaxic coordinates AP = −4.0 mm, ML = +/− 2.5 mm and DV = −1.5 mm to −4.5 mm relative to the Bregma. In addition to tungsten wires, we used copper wires (0.28 mm diameter) to measure the muscular activity (electromyogram EMG) in the neck muscles. Before each surgery, the relative position of each recording site on the electrode (8 recording sites per electrode) was identified by measuring the impedance change, while lowering the electrodes in saline solution (Na-Cl 0.9%). The actual design was based on handmade electrodes with minimal spacing of 500 microns between recording sites and a maximal number of 8 recording sites per electrode. This allows to observe the characteristic phase reversal between the superficial and deep layers of the dorsal hippocampus, but not to quantify the cross-frequency coupling as with linear probes. Additionally, the surgical procedure is complex and there is variability between targeted structure and actual electrode position due to brain tissue movement (swelling) both during and after the surgery. See Supplementary Fig. 3 for further details.

**Electrode implantation verification**. Electrodes sites' locations were verified post mortem via histology to reconstruct the tract of electrode bundles in the tissue. Each rat was euthanized and perfused with paraformaldehyde 4% to preserve the brains. Each brain was then cut using a vibratome to make 100μm-thick slices. The slices were then contrasted using hematoxyline/eosine coloration and scanned using a nanoscan. We then compared the slices with plates from Paxinos and defined the trajectory of implantation using the marks left by the electrode in the brain. Knowing the distances between the recording points, we could then define their position.

**Recording sessions**. After a recovery week following the surgical procedure, the animals were fit to undergo data acquisition. After applying a generous amount of centrifuged ultrasonic gel, the ultrasonic probe was put in place using a magnetic probe holder (home-made 3D designed and printed) and the headstage for LFP recordings was plugged onto the connector. The animal was then placed inside a box under an infrared camera (to monitor the behavior) and the data acquisition started. The ultrasonic probe was placed randomly across the day and its position was changed after 2–3 30 min acquisitions. A typical recording session scanned 4 different brain plans and lasted ~6 h with different breaks. All the probe and headstage positioning and moving was realized without having to put the animal under anesthesia. At the end of the recording session, the window is cleaned from excess ultrasonic gel and the animal is replaced in its home-cage.

**LFP acquisition**. LFP, electromyogram (EMG) and accelerometer (ACC) signals and video were monitored continuously from video-EEG device for offline processing. Intracranial electrode signals were fed through a Blackrock Cereplex System using the Cereplex Direct Software Suite (version 7.0.6.0) developed by Blackrock Microsystems (Salt Lake City, UT, USA), together with a synchronization signal from the ultrasound scanner. LFP signals were pre-amplified and digitized onto the animal's head which prevent artifacts originating from cable movement. A regular amplifier was used, and no additional electronic circuit for artifact suppression was necessary. A large bandwidth amplifier was used, which can record local field potentials in all physiological bands (LFP, 0.1–2 kHz). The spatial resolution of LFPs ranges from 250 μm to a few mm radius.

**Ultrasound Acquisition**. Vascular images were obtained via the ultrafast compound Doppler imaging technique. The probe was driven by a fully programmable GPU-based ultrafast ultrasound scanner Verasonics (Kirkland, USA), relying on 24-Gb RAM memory. We continuously acquired $N = 4500$ Doppler ultrasound images at 2.5 Hz frame rate for 30 min straight. Each Doppler frame is obtained using the accumulation of 200 successive compound plane-wave frames, each compounded frame corresponding to a coherent summation of beamformed complex in phase/quadrature (IQ) images obtained from the insonification of the medium with a set of successive plane waves with specific tilting angles. Given the tradeoff between frame rate, resolution and imaging speed, a multi plane-wave compounding using eight 2°-apart angles of insonification (from −7° to +7°) was chosen. As a result, the pulse repetition frequency of the plane wave transmissions was 4000 Hz. To discriminate blood signals from tissue clutter, the ultrafast compound Doppler frame stack was filtered, removing the 60 first components of the singular value decomposition, which optimally exploited the spatiotemporal dynamics of the full Doppler film for clutter rejection, largely outperforming conventional clutter rejection filters used in Doppler ultrasound. All these parameters taken together we obtain a Doppler movie with 100 ms order temporal resolution and 100 μm order spatial resolution.

**LFP analysis**. All analysis was performed in MATLAB (version R2017b, Mathworks, USA). NPMK package (version 4.5.3.0) developed by Blackrock Microsystems was used to import the raw LFP data into MATLAB. EEG was collected and low-pass filtered under 250 Hz. This, together with direct amplification onto the animal's head via the INTAN chip from the Blackrock system, allows for quality and artifact-free LFP recording in the hippocampus and the thalamus in the freely moving animal. EEG was then band-pass filtered in typical frequency bands including delta (1–4 Hz), theta (4–10 Hz), beta (10–20 Hz), low-gamma

(20–50 Hz), mid-gamma (50–100 Hz), upper mid-gamma (80–120 Hz), high-gamma (100–150 Hz), upper high-gamma (130–180 Hz) and ripple (150–250 Hz). This division has been thoroughly described and proven to be functionally relevant for hippocampal electrographic recordings[65].

**Sleep scoring**. Signals from the EMG, ACC and LFP recordings were used to perform the sleep scoring. It allowed us to separate four different vigilance states: active wake (AW), quiet cake (QW), REM sleep (REMS) and non-REM sleep (NREMS). Each of these states is characterized by a combination of indexes drawn from the recorded signals. We computed the power of the EMG signal (as used in[66]), the ratio theta/delta of the LFP signal and then manually thresholded these two parameters and the accelerometer signal to obtain indexes made of zeros and ones. The four different vigilance states are defined thus: AW = EMG 1 + ACC 1 + ratio 0/1; QW = EMG 1 + ACC 0 + ratio 0/1; REM = EMG 0 + ACC 0 + ratio 1; NREMS = EMG 0 + ACC 0 + ratio 0. The IR video recording was used from time to time to confirm the sleep scoring.

**CBV maps and spatial averaging and activation threshold**. Previous studies from our group have shown that the fUS signal tightly relates to neuronal activity and microscopic single-vessel hemodynamics. In order to build the CBV maps from the raw back-scattered echoes, radiofrequency (RF) signals are delayed and summed to form IQ matrices through a process known as beamforming. Theses matrices are then decomposed via Singular Value Decomposition (SVD) to decouple slow movements due to pulsatility and tissue motion from fast movements due to echogenic particles crossing a voxel during a full cardiac cycle (200 ms). Importantly, Power Doppler images are computed by taking the power of the full Doppler spectrum, including a range of speeds in large and smaller vessels, with a typical inferior bound of 2–5 mm/s in axial velocity. This gives a signal proportional to the number of echogenic particles that have crossed a single voxel during 200 ms (with a sufficient axial velocity) which is a good estimate of local cerebral blood flow (CBF). We thus can build Doppler movies with a sampling frequency of 2.5 Hz, which can even be increased if needed up to the pulse repetition frequency (here 500 Hz) through the use of a temporal sliding window. To derive CBV maps from the raw Doppler movies, we performed voxel-wise normalization from a baseline period: depending on the analysis done afterwards, we either used 2 min of quiet wake, 2 min of active wake, or the 20 s preceding the onset of a REM episode. We extracted the distribution for each voxel during this baseline period and computed a mean value, leading to one value for each voxel of the image. To derive a signal similar to ΔF/F in fluorescence microscopy, we subtracted the mean and divided by the mean for each voxel in the Doppler movie. This allowed normalization and rescaling of ultrasound data, yielding to an expression in terms of percent of variation relative to baseline (CBV % change). Each voxel was normalized independently before performing spatial averaging. The activation threshold used to classify pixels as active or inactive during REMS episodes, was set as μAW + n*σAW, where μAW and μAW are respectively the mean and standard deviation of pixel CBV signal after normalization (expressed as %CBV change). $n$ was set to 1. This means that a pixel was considered 'active' when the difference between its value and the mean of the AW distribution was greater than one standard-deviation of the AW distribution. This definition imposed that pixel activity outmatch active wake levels to be considered 'active' which is significantly more stringent than using the mere QW distribution.

**Atlas registration and identification of vascular structures**. Coronal and sagittal motored scans allowed for the registration of the recordings to two-dimensional sections from the Paxinos atlas[67] using anatomical landmarks, such as cortex edges, hippocampus outer shape and large vessels below brain surface as a reference. We performed manual scaling and rotation along each of the 3 dimensions to recover the most probable registration. Once performed, regions of interest were extracted using binary masks. This process allowed us to derive vascular activity in 259 brain regions.

An automatic registration software developed in our lab is currently available for the mouse's brain through the co-registration of a vascular atlas on the Allen Brain Institute's atlas[68]. This fully automatic approach based on the cerebral vascular print recognition demonstrated a subpixel (<100 μm) precision outperforming the manual registration performed by highly skilled neuroscientists (<200 μm). Such Approach will be available in the near future for rats, using recently published vascular and MRI atlases in rats allowing for infra-pixel precision.

As for the identification of the vascular structures, we used both the result of our manual registration and the few vascular atlases currently available in rats and mice[69,70]. We also performed Ultrasound Localisation Microscopy in a particular sagittal plane to distinguish between two close veins. This technique enables, through the detection and tracking of individual micro-bubbles injected intravenously that constitute contrast agents[71], to visualize the vasculature with a higher sensitivity and contrast, but also to measure the direction and speed of blood flow at a microscopic scale in both rodents[71] and human[72].

**Cross-correlation analysis**. We used two-sided Pearson's cross-correlation score to quantify the association between REM episodes and the CBV activations in brain

regions, or between brain regions, or between LFP signals and CBV activations in brain regions. To do so, we performed the cross-correlation computation on a large temporal window, depending on the couple analyzed. Regarding the CBV activations, we either used pixels (Fig. 5c–e) or averaged values over brain regions or structures.

**LFP-CBV correlation analysis**. To assess the association between LFP events and CBV variables, we searched for correlations between each possible combination of LFP band-pass filtered signals and regional CBV variables. As neurovascular processes are not instantaneous, we considered possible delays between electrographic and vascular signals and thus computed cross-correlations functions between the two signals for any LFP-CBV pair and any lag in a given time window (−1.0 s to 5.0 s). We performed this analysis over pixel and regional variables, but only regional variables allowed for statistical comparison across recordings.

**Statistics and reproducibility**. All statistics are given as +/− standard error of the mean unless stated otherwise. All data shown in this paper have been acquired on $N = 3$ animals including 72 recordings and a total of 577 REMS episodes and an additional 5 rats and 12 recordings solely for Fig. 4. Bar plots in Fig. 1c are computed by the distribution of mean CBV values per recording for every region and deriving a global mean +/− sem for each region. Box plots in Fig. 2d are computed by pooling each frame within either the REM-PHASIC or REM-tonic group and computing a spatial recruitment ratio (% active pixels) within each region. Statistical testing in Fig. 2e was conducted under the null hypothesis 'no difference between the mean values observed in the two groups' via Wilcoxon signed rank test (n.s. non-significant: $P > .05$, *$P < .05$, **$P < .01$, ***$P < .001$). Exact values are provided as a separate file. Correlograms in Fig. 3b are derived by averaging the regional correlogram obtained for each recording using either seed-REM, seed-REM-tonic or seed-REM-phasic for each region. A mean correlogram is then extracted from which we derived a mean + /1 std interval. Bar plot in Fig. 3d are computed similarly as in Fig. 1c (one value per recording). Data in Supplementary Fig. 3 are obtained by pooling the mean regional value for each temporal frame within either of the 4 groups quiet wake QW, active wake AW, NREMS non-REM sleep, REMS REM sleep. Box plots show respectively the minimal (left whisker), 25% quartile (left notch), median (middle bar), 75% quartile (left notch) and maximal value (right whisker).

**Reporting summary**. Further information on research design is available in the Nature Portfolio Reporting Summary linked to this article.

## Data availability
Source data values underlying Figs. 1c, 2d, e, 3b, d are available in Supplementary Data 3. All other data supporting the findings of this study are available from the corresponding authors upon reasonable request.

## Code availability
Custom codes used for the collection and analysis of fUS/LFP/video data used in this study are protected by Inserm and can only be shared upon request, with the written agreement of Inserm.

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

## Acknowledgements

The authors would like to thank F. Segura, M. Nouhoum, H. Serroune, A. Bertolo for their help in the different steps of the research leading to these results. The research leading to these results has received funding from the AXA Research Fund under the chair "New hopes in medical imaging with ultrasound". The authors would also like to thank Dr. Isabelle Arnulf and Dr. Karim Benchenane for their fruitful scientific discussions and advice.

## Author contributions

A.B. and M.T. designed the experiment. A.B. designed the electrodes and performed the surgeries. M.M. crafted the electrodes, performed the training and recording sessions. A.B. programmed the software for multimodal data visualization and analysis. A.B. and M.M. analyzed the behavioral, electrographic data, and ultrasound data. M.M. and S.P. performed ULM experiments and identified vascular structures. All authors discussed multimodal analysis. All authors wrote the paper.

## Competing interests

The Authors declare the following competing interests: M.T. is co-founder and shareholder in the ICONEUS company. All other authors declare no competing interests.
