## [Peer Review File · Communications Biology]

Reviewers' comments:

Reviewer #1 (Remarks to the Author):

Brief summary of the manuscript

Local changes of cerebral blood volume (CBV) in the whole brain of rats during REM sleep were measured plane-by-plane sequentially using functional ultrasound imaging (fUS). Authors found widespread, sustained increase in CBV in the whole brain during both tonic and phasic REM sleep, whereas transient, widespread CBV increase in cortical structures was associated with phasic REM sleep. Strikingly, the amygdala was disconnected from global activation in the rest of the brain during REM sleep. Additionally, fUS was able to identify vascular structure and arterial blood supply during REM sleep.

Overall impression of the work

First of all, I commend the authors for conducting this technically challenging, labor-intensive project. This study was conducted by the group who used ultrasound technology to develop a new, full-fledged brain imaging technique with very high spatio-temporal resolution (Deffieux, Demené and Tanter, 2021).

Global dissociation of the amygdala from hyperemia in the rest of the brain during REM sleep is a novel finding, to my knowledge. The association of transient synchronous hyperemia in the cortex with phasic REM sleep is also of interest. These are thought provoking findings.

fUS imaging is a powerful, promising tool to study interaction between large scale brain networks during REM sleep. In particular, fUS can produce valuable data about the neural correlates of rapid eye movements (REMs). REMs are indexed to seeing things in the dreaming mind and hierarchical processing of visual information by the dreaming brain (Hong, Fallon, Friston, Harris, 2018, Rapid eye movements in sleep furnish a unique probe into consciousness <https://www.frontiersin.org/articles/10.3389/fpsyg.2018.02087/full>). Notably, solid evidences indicate that the brain sees things the same way in dreaming and while awake. fUS study of REM sleep should be encouraged as it will provide valuable data about dreaming and waking consciousness.

There may, however, be a better explanation for the isolation of the amygdala during REM sleep than the one provided by the authors, as I will discuss in the following section.

Specific comments, with recommendations for addressing each comment

The authors interpret "global dissociation of amygdala from the rest of the brain during REM sleep" as follows: "amygdala ... may be linked to the creation of dream content during [REM sleep]" (line 32 & 33, in abstract). There may be a better explanation. fMRI evidence indicates that generation of dreaming (or waking) consciousness is linked to widespread brain activation (Hong, Fallon, Friston, Harris, 2018). Since the amygdala is disconnected from widespread brain activation during REM sleep, the amygdala would not be able to contribute to the creation of dream content.

Dissociation of the amygdala may instead be a mechanism that prevents generation of negative emotions, thereby precluding an interruption to the energy-expensive process of generating consciousness. The authors observed synchronous, global hyperemia during phasic REM sleep. Global availability of incoming sensory information to multiple brain systems may be essential for conscious experience (Dehaene and Changeux, 2011; Metzinger, 2003). Dissociation of the amygdala indicates that information generated in the amygdala is not available to the rest of the brain during REM sleep.

Furthermore, interconnectivity among amygdala nuclei as well as activity in the amygdala itself are reduced during REM sleep. Dissociation and suppression of the amygdala may protect the energy-expensive generation of dream-consciousness from interruption by precluding the intrusion of fear into dreaming consciousness. Failure to suppress and isolate the amygdala may lead to nightmares and disruption of sleep, and may play a role in development of PTSD.

Line 319-323

“Indeed, a higher vascular activity in amygdala was already found correlated with REMS using positron emission tomography in humans (Maquet et al., 1996). Interestingly, our results also indicate that this amygdala hyperemia presents a lower correlation with the rest of the brain regions, depicting a global dissociation of the amygdala from other brain regions.”

The O-15 PET findings of increased amygdala activity during REM sleep are different from the fUS findings. Authors found relatively low hyperemia in the amygdala during REM sleep.

Line 321-322

“Interestingly, our results also indicate that this amygdala hyperemia presents a lower correlation with the rest of the brain regions”

The meaning of this sentence is not clear. Does “this amygdala hyperemia” mean the O-15 PET finding? fUS found relatively low-level amygdala hyperemia. Additionally, regardless of increase or decrease in amygdala hyperemia, its correlation with the rest of the brain regions can be lower. They are separate, not causally linked.

Authors reported widespread increase in cortical blood flow with transient peaks (‘vascular surge’) (measured by fUS) during phasic REM sleep in rats, reproducing the results of Bergel et al. (2018). (Phasic REM sleep is the part of REM sleep with rapid eye movements. A substantial part of REM sleep is without rapid eye movements, and this part is called tonic REM sleep.) This finding is consistent with an earlier finding, that widespread activation (measured by event-related fMRI) is time-locked to rapid eye movements in sleeping human subjects (Hong et al. 2009 <https://pubmed.ncbi.nlm.nih.gov/18972392/>). The time course of brain-wide massive CBV spikes lasting 5 to 30 s during phasic REM sleep (‘vascular surge’) in a fUS study (Bergel et al., 2018, Figure 1b) is similar to the actual time course of fMRI BOLD signal changes as well as the expected time course of the hemodynamic model in an “event”-related fMRI study—“event” being rapid eye movements (Hong et al., 2009, Figure 1a). It may be that the ‘vascular surge’ during phasic REM sleep in rats is time-locked to rapid eye movements in phasic REM sleep. If the authors agree with this interpretation, the introduction and discussion should be changed accordingly.

The authors should consider identifying and timing rapid eye movements (REMs) in their future studies, perhaps with the infrared eye tracker they used in their previous study (Dizeux et al., 2019). First, rapid eye movements are the hallmark of REM sleep and the key component of phasic REM sleep. Thus, it would be ideal for a study of REM sleep (in particular, a study of phasic REM sleep) to include identification of rapid eye movements. For example, Shein-Idelson et al. (2016) quantified rapid eye movements of Australian dragons by rendering video recordings of closed eyes to computerized analysis. Second, it will enable better comparisons and synthesis with other studies on animal and human subjects. Findings can be synthesized across species and also across life spans (rapid eye movements can be easily identified in human infants and adults using video monitoring). Third, I am confident that timing rapid eye movements in future fUS studies will provide valuable data. Studying CBV changes time-locked to REMs will contribute to the science of consciousness, in particular, neurodevelopment of visual perception (Hong, Fallon, Friston, Harris, 2018). REMs are simultaneously indexed to seeing things in the dreaming mind as well as to hierarchical processing of visual information in the dreaming brain, providing a valuable probe into the link between the mind and the brain. The brain sees things the same way in dreaming and while awake. Practically, rapid eye movements are straightforward, temporally precise events, and time series analyses of rapid eye movements have statistical efficiency. The authors may consider adding the timing of rapid eye movements as a future direction for research in their manuscript.

This group studied brain activity in human newborns using trans-fontanel fUS imaging (Demene, et al., 2017; Baranger, et al., 2021). The marked preponderance of REM sleep in the last trimester of pregnancy (fetuses are in REM sleep for almost the whole day) and in infancy (neonates are in REM sleep for up to 50% of sleep at birth) indicates the important role of REM sleep in neurodevelopment (Hobson, 2009). “As a natural, task-free probe, rapid eye movements in sleep could be used in non-compliant subjects, including infants and animals. In short, REMs constitute a promising probe to study the ontogenetic and phylogenetic development of consciousness” (Hong, Fallon, Friston, Harris, 2018).

Additionally, event-related fMRI findings of rapid eye movements (Hong et al., 2009) will help fUS researchers of REM sleep to identify the most notable brain structures on which to focus in their fUS studies, namely, V1, thalamic reticular nucleus, claustrum, cholinergic basal nucleus, retrosplenial cortex (right and left separately). REM-locked activation is widespread, but REM-locked activation peaks (which were identified after raising the statistical threshold to corrected $P < 0.00005$) are clearly localized in the structures listed above (Hong et al., 2009). Studying these structures with fUS that has high sensitivity and 10-100 ms temporal resolution (Deffieux et al., 2021)—enabling event-related assessment of the top-down directional propagation of signals in hierarchical processing after only a single trial of visual tasks (Dizeux et al., 2019)—will no doubt expand our knowledge of dreaming and waking consciousness. fUS studies employing rapid eye movement events as a probe will advance exploration of brain network dynamics unperturbed by external stimuli (i.e., while much of the external sensory input to the brain is blocked during REM sleep). The strategy of identifying notable brain structures to study with fMRI (which offers simultaneous measurement of the whole brain rather than plane-by-plane measurements) before single unit recording worked well to elucidate face recognition (Chang and Tsao, 2017).

It was reported last month that the relatively small areas exempt from widespread REM-locked brain activation were restricted to the default mode network (Hong, Fallon, Friston, 2021). I would like to know if the authors observed in the default mode network a similar pattern of attenuation of the brain-wide ‘vascular surge’. In particular, I would like to know if they observed attenuation of hyperemia in the left retrosplenial cortex during phasic REM sleep, as we observed in our fMRI study. Ferrier et al. (2020) reported sensory task-induced bilateral CBV decrease in granular retrosplenial cortex (RSG) (greater decrease in Lt > Rt). It is interesting to note that a solitary small area corresponding to the left RSG was identified as the 5th component by ICA.

Both fUS and fMRI measure hemodynamic changes. This fUS study of REM sleep observed vasodilation in the whole brain. This may illuminate our earlier observation of REM-locked vasodilation, indicated by a robust periventricular fMRI BOLD signal decrease time-locked to rapid eye movements (Hong, Fallon, Friston, 2021). It is unlikely that REM-locked vasodilation occurs only in the periventricular areas in humans. Rather, it may be that fMRI (due to its nature) can detect vasodilation only at the water-brain tissue border (because of partial volume effect) even though vasodilation occurs brain-wide in humans as well as in rats.

Lines 102 – 105 “yet this hyperemic activity seems to be extremely energy consuming. We assume that if such energy-demanding activity has been maintained across evolution, it must have some important role for the survival of the animal, which does not seem to have been found yet.”

Lines 258 – 259 “Such hyperemic activity might be physiologically important as it was kept throughout evolution, despite its energy consumption.”

Consciousness is constructive and inferential. Generation of consciousness is linked to widespread brain activation, which is energy demanding, although the brain strives for energy efficiency (Hong, Fallon, Friston, Harris, 2018). The crux of consciousness generation—the world-model, with the self-model at its center—can be viewed as “a wonderfully efficient control device” (Metzinger, 2009) enabling interaction with the world, which is essential for the survival of individuals and of the species. If in fact this global hyperemic activity is linked to consciousness generation, which is essential for

survival, this would justify its extremely high energy consumption.

Lines 67 and 69

The authors may consider adding the following paper in the reference: Hobson, J. A. (2009). REM sleep and dreaming: towards a theory of protoconsciousness. *Nat. Rev. Neurosci.* 10, 803–813. doi: 10.1038/nrn2716

“or the brain maturation specifically during early life REMS (Boyce et al., 2016; Hobson, 2009) and more precisely to aid sensorimotor system development (Hobson, 2009) through muscle twitches (Blumberg et al., 2013).”

Related to this neurodevelopmental function of REM sleep, Hobson proposed that REM sleep provides off-line practice for walking in utero, before we actually learn to walk after birth (Hobson 2017, cited in Hong et al., 2018).

[Hobson, A. (2017). *Conscious States: The AIM Model of Waking, Sleeping, and Dreaming*. Scotts Valley, CA: Create Space.]

Rats were used for the study, but ‘rat’ was not mentioned in the abstract.

CBV was not defined. I assume that CBV stands for cerebral blood volume.

Figure 1 legend. “color: regional CBV traces” What color?

Figure 3D Gangliacampus -> Ganglia campus

Table 1 -> Supplementary Table 1

Reviewer #2 (Remarks to the Author):

The current work uses functional ultrasound imaging to examine and characterize brain-wide vascular responses to different stages of sleep in rats. The work extends previous work by researchers who are well-versed in both functional ultrasound and sleep research. While there are a few key take away points from the work, such as the dissociation of amygdalar activity from the rest of the brain during REMS, much of this work also overlaps with one of the authors’ previous papers. While a valiant effort was put into this work, the analysis is fairly simple and shallow, and the authors do not present many significant and/or well-supported results. In its current state, I do not think this manuscript can be accepted for publication and needs significant work in terms of both the analysis and narrative. For this reason I am recommending rejection of this manuscript. I am sorry that I cannot be more supportive at this time, but wish the authors the best of luck in future submissions. Please see my detailed points below:

1) I find the overall narrative of the current work in terms of the role of REMS in emotion regulation a bit grandiose. None of the experiments described in this work indicate any involvement of emotions, neither via observation nor via stimulus delivery. I feel that the need to elucidate REMS is a strong enough motivation for the current work, which also builds nicely off a previous study performed by many of the same authors. I wonder if the authors can simply expand this line of thought without incorporating emotions.

2) The information provided in Figures 1 and 2 largely overlaps with that in one of the authors’ previous works (Bergel et al. 2018 *Nature Communications*). From what I can tell, it seems that the main difference between Figures 1 and 2 in these two manuscripts is that the current ones include information from a larger field-of-view. In this sense, I mean that Bergel et al. 2018 only examined a single plane and the current work examined multiple planes (coronal and sagittal), and therefore include more brain regions. While this additional information in theory represents a more comprehensive evaluation of these vigilance states, the analysis performed does not seem to teach

the reader anything new that could not have been learned from the previous paper. The authors should explain the results from these figures in a clearer way and specifically describe about how this information deviates from that in the previous work.

3) I find the comparison between the tonic and phasic activation problematic for a few reasons. In general, the argument that various cortical structures are more correlated with phasic rather than tonic REMS is weakly supported by the results presented. To more clearly support this claim, I would expect at least a difference map between the two correlation maps presented in Figure 3C. Currently the correlation maps for tonic and phasic REMS look very similar and one has to look quite closely to see the differences pointed out by the authors. A difference map would clearly show instances of higher/lower correlation between the two conditions.

In addition to this, I feel that a simple correlation analysis is not the strongest way to show the difference desired by the authors. A general linear model (GLM) analysis would more comprehensively and with greater statistical power evaluate potential differences in tonic vs phasic activation. This brings me to an additional point of how the tonic and phasic regressors (or seeds) were constructed. I don't understand why the phasic regressor would also contain tonic information, and would rather think that two binary regressors of only zeros and ones (1 phasic REMS, 0 otherwise and 1 tonic REMS, 0 otherwise) would be more appropriate. This may also be necessary for a GLM analysis as the current regressor designs (as depicted in Figure 3A) seem highly collinear and could lead to unstable GLM results. Nevertheless, with orthogonal regressors (as I describe above), the authors should be able to obtain stable tonic and phasic REMS maps and further be able to obtain a contrast map of tonic vs phasic REMS. I strongly encourage the authors to explore this option.

4) The amygdalar phenomenon is quite interesting and seems convincing in terms of the connectivity analyses in Figure 5 A, C and D. However, the time series information presented in Figure 5B doesn't portray as strong of an argument. It does appear that the amygdalar sub-regions do deviate from the whole brain trace, but only slightly. To determine how meaningful this deviation is, it would be useful to also include a time series trace from a region(s) that is highly correlated with the whole brain signal. In this sense, the reader could clearly see by eye what relatively high and low correlated activity looks like.

Furthermore, the authors present the results of very small amygdalar sub-regions, but do not discuss how reliable the segmentation of these areas are. While it is generally understood the functional ultrasound exhibits a high spatial resolution, I cannot find the pitch of the probe stated anywhere in the methods. This lack of specified resolution, combined with no scale bars shown in the images in Figure 5C-E make it difficult to assess whether it is reasonable that functional ultrasound can resolve structures at the spatial scale being claimed. I would further assume that errors from the manual registration process and variability in window width/quality could further complicate the accurate identification of these regions. It would be useful if the authors were to clarify these important pieces of information.

Global dissociation of the amygdala from the rest of the brain during REM sleep

M. Matei et al.

Review received : 28/02/2022

Review expected : 28/08/2022

Reviewers' comments:

Reviewer #1 (Remarks to the Author):

Brief summary of the manuscript

Local changes of cerebral blood volume (CBV) in the whole brain of rats during REM sleep were measured plane-by-plane sequentially using functional ultrasound imaging (fUS). Authors found widespread, sustained increase in CBV in the whole brain during both tonic and phasic REM sleep, whereas transient, widespread CBV increase in cortical structures was associated with phasic REM sleep. Strikingly, the amygdala was disconnected from global activation in the rest of the brain during REM sleep. Additionally, fUS was able to identify vascular structure and arterial blood supply during REM sleep.

Overall impression of the work

First of all, I commend the authors for conducting this technically challenging, labor-intensive project. This study was conducted by the group who used ultrasound technology to develop a new, full-fledged brain imaging technique with very high spatio-temporal resolution (Deffieux, Demené and Tanter, 2021). Global dissociation of the amygdala from hyperemia in the rest of the brain during REM sleep is a novel finding, to my knowledge. The association of transient synchronous hyperemia in the cortex with phasic REM sleep is also of interest. These are thought provoking findings.

We deeply thank reviewer #1 for this very positive and supportive comments towards our study and findings.

fUS imaging is a powerful, promising tool to study interaction between large scale brain networks during REM sleep. In particular, fUS can produce valuable data about the neural correlates of rapid eye movements (REMs). REMs are indexed to seeing things in the dreaming mind and hierarchical processing of visual information by the dreaming brain (Hong, Fallon, Friston, Harris, 2018, Rapid eye movements in sleep furnish a unique probe into consciousness <https://www.frontiersin.org/articles/10.3389/fpsyg.2018.02087/full>). Notably, solid evidence indicates that the brain sees things the same way in dreaming and while awake. fUS study of REM sleep should be encouraged as it will provide valuable data about dreaming and waking consciousness.

We thank reviewer #1 for this positive comment and fully agree that fUS imaging study of REM sleep should be encouraged and could provide a lot of new insights in the field of dreaming and waking consciousness.

There may, however, be a better explanation for the isolation of the amygdala during REM sleep than the one provided by the authors, as I will discuss in the following section.

We thank reviewer #1 for his proposed interpretation for the isolation of the amygdala. We added this explanation in the discussion part and reformatted the introduction in order to take into account this proposition. We also deeply thank reviewer #1 for his major help in improving the manuscript and the detailed list of proposed references we added in the manuscript.

Specific comments, with recommendations for addressing each comment

The authors interpret “global dissociation of amygdala from the rest of the brain during REM sleep” as follows: “amygdala ... may be linked to the creation of dream content during [REM sleep]” (line 32 & 33, in abstract). There may be a better explanation. fMRI evidence indicates that generation of dreaming (or waking) consciousness is linked to widespread brain activation (Hong, Fallon, Friston, Harris, 2018). Since the amygdala is disconnected from widespread brain activation during REM sleep, the amygdala would not be able to contribute to the creation of dream content.

Dissociation of the amygdala may instead be a mechanism that prevents generation of negative emotions, thereby precluding an interruption to the energy-expensive process of generating consciousness. The authors observed synchronous, global hyperemia during phasic REM sleep. Global availability of incoming sensory information to multiple brain systems may be essential for conscious experience (Dehaene and Changeux, 2011; Metzinger, 2003). Dissociation of the amygdala indicates that information generated in the amygdala is not available to the rest of the brain during REM sleep. Furthermore, interconnectivity among amygdala nuclei as well as activity in the amygdala itself are reduced during REM sleep. Dissociation and suppression of the amygdala may protect the energy-expensive generation of dream-consciousness from interruption by precluding the intrusion of fear into dreaming consciousness. Failure to suppress and isolate the amygdala may lead to nightmares and disruption of sleep, and may play a role in development of PTSD.

Although we don't have enough results in our work to confirm this, this is an extremely interesting interpretation of the results presented in the manuscript and we deeply thank you for pointing it out. We understand here that our explanations about the role of amygdala in the treatment of negative emotions were not clear enough and should be incorporated in this more general interpretation.

We fully agree that, in light of the fact that the generation of dreaming consciousness is linked to widespread brain activation, the dissociation of amygdala favors the assumption that it should not contribute to the dream content but rather would require it to be disconnected from the rest of the brain. Your interpretation of the role of amygdala in interrupting the REM sleep process due to the intrusion of fear into dreaming consciousness is very attractive. We modified the discussion section in order to introduce this interpretation.

Line 319-323

“Indeed, a higher vascular activity in amygdala was already found correlated with REMS using positron emission tomography in humans (Maquet et al., 1996). Interestingly, our results also indicate that this amygdala hyperemia presents a lower correlation with the rest of the brain regions, depicting a global dissociation of the amygdala from other brain regions.”

The O-15 PET findings of increased amygdala activity during REM sleep are different from the fUS findings. Authors found relatively low hyperemia in the amygdala during REM sleep.

We agree that relatively low hyperemia in the amygdala was found using fUS imaging. We have thus modified the sentence “Although a relatively low hyperemia in the amygdala was found during REM sleep using fUS imaging, our results show that the amygdala vascular activity presents a lower correlation with the rest of the brain regions, depicting a global dissociation of the amygdala from other brain regions.”

A possible explanation to reconcile the discrepancies in our findings and the O-15 PET findings is the difference of models used (human versus rat), the parameters measured by the two techniques (oxygen metabolism versus cerebral blood volume) and the fact that the disconnection is strongest in the basal/cortical/lateral amygdala than in the central amygdala (Figure 3).

Line 321-322

“Interestingly, our results also indicate that this amygdala hyperemia presents a lower correlation with the rest of the brain regions”. The meaning of this sentence is not clear. Does “this amygdala hyperemia” mean the O-15 PET finding? fUS found relatively low-level amygdala hyperemia. Additionally, regardless of increase or decrease in amygdala hyperemia, its correlation with the rest of the brain regions can be lower. They are separate, not causally linked.

We fully agree that the sentence was not clear and it was rewritten: “Although a relatively low hyperemia in the amygdala was found during REM sleep using fUS imaging, our results show that the amygdala vascular activity presents a lower correlation with these in the rest of brain regions, depicting a global dissociation of the amygdala from other brain regions.”

Authors reported widespread increase in cortical blood flow with transient peaks (‘vascular surge’) (measured by fUS) during phasic REM sleep in rats, reproducing the results of Bergel et al. (2018). (Phasic REM sleep is the part of REM sleep with rapid eye movements. A substantial part of REM sleep is without rapid eye movements, and this part is called tonic REM sleep.) This finding is consistent with an earlier finding, that widespread activation (measured by event-related fMRI) is time-locked to rapid eye movements in sleeping human subjects (Hong et al. 2009 <https://pubmed.ncbi.nlm.nih.gov/18972392/>). The time course of brain-wide massive CBV spikes lasting 5 to 30 s during phasic REM sleep (‘vascular surge’) in a fUS study (Bergel et al., 2018, Figure 1b) is similar to the actual time course of fMRI BOLD signal changes as well as the expected time course of the hemodynamic model in an “event”-related fMRI study—“event” being rapid eye movements (Hong et al., 2009, Figure 1a). It may be that the ‘vascular surge’ during phasic REM sleep in rats is time-locked to rapid eye movements in phasic REM sleep. If the authors agree with this interpretation, the introduction and discussion should be changed accordingly.

This is an excellent comment and we thank the reviewer for this suggestion. We added these important references and modified the introduction and discussion accordingly. To expand on this point, vascular surges are time-locked to hippocampal theta and gamma bursts during phasic REM sleep (they trail by 1 to 2 seconds depending on the region) (Bergel et al. 2018). These theta/gamma bursts of activity have been shown to precede fluctuations in arterial pressure (Sei and Morita, 1996) and it is possible that they are time-locked to the other components of phasic REM sleep, such as muscle twitches, REM, whisking and penile erections, though this remains to be demonstrated. This is an important line of research for the future and we thank the reviewer for pointing this out.

The authors should consider identifying and timing rapid eye movements (REMs) in their future studies, perhaps with the infrared eye tracker they used in their previous study (Dizeux et al., 2019). First, rapid eye movements are the hallmark of REM sleep and the key component of phasic REM sleep. Thus, it would be ideal for a study of REM sleep (in particular, a study of phasic REM sleep) to include identification of rapid eye movements. For example, Shein-Idelson et al. (2016) quantified rapid eye movements of Australian dragons by rendering video recordings of closed eyes to computerized analysis. Second, it will enable better comparisons and synthesis with other studies on animal and human subjects. Findings can be synthesized across species and also across life spans (rapid eye movements can be easily identified in human infants and adults using video monitoring). Third, I am confident that timing rapid eye movements in future fUS studies will provide valuable data. Studying CBV changes time-locked to REMs will contribute to the science of consciousness, in particular, neurodevelopment of visual perception (Hong, Fallon, Friston, Harris, 2018). REMs are simultaneously indexed to seeing things in the dreaming mind as well as to hierarchical processing of visual information in the dreaming brain, providing a valuable probe into the link between the mind and the brain. The brain sees things the same way in dreaming and while awake. Practically, rapid eye movements are straightforward, temporally precise events, and time series analyses of rapid eye movements have statistical efficiency. The authors may consider adding the timing of rapid eye movements as a future direction for research in their manuscript.

The reviewer is fully right and we definitely would like to add rapid eye movements tracking in the future studies with fUS. We added this perspective in the discussion and thank the reviewer again for these comments strongly helping us to improve the manuscript. Unfortunately, in this set of experiments, the camera was placed well-above the recording box, with the intention to cover the full spatial extent of the box, and rapid-eye-movements are not available from this viewpoint. We thus intend to add close-up cameras to record at ground level and be able to track rapid-eye-movement and include them in further analyses of REM sleep with functional ultrasound activity.

This group studied brain activity in human newborns using trans-fontanel fUS imaging (Demene, et al., 2017; Baranger, et al., 2021). The marked preponderance of REM sleep in the last trimester of pregnancy (fetuses are in REM sleep for almost the whole day) and in infancy (neonates are in REM sleep for up to 50% of sleep at birth) indicates the important role of REM sleep in neurodevelopment (Hobson, 2009). “As a natural, task-free probe, rapid eye movements in sleep could be used in non-compliant subjects, including infants and animals. In short, REMs constitute a promising probe to study the ontogenetic and phylogenetic development of consciousness” (Hong, Fallon, Friston, Harris, 2018).

We fully agree that tracking of REMs in neonates would also be highly valuable in conjunction with fUS imaging. Interestingly, active sleep in neonates is also associated with slow and large-amplitude fluctuations in the vascular system (akin to vascular surges), which are very similar to those found with fUS imaging in rats. It would be interesting to correlate rapid-eye-movements with these vascular surges found in neonates during active sleep (Demene et al 2017).

Additionally, event-related fMRI findings of rapid eye movements (Hong et al., 2009) will help fUS researchers of REM sleep to identify the most notable brain structures on which to focus in their fUS studies, namely, V1, thalamic reticular nucleus, claustrum, cholinergic basal nucleus, retrosplenial cortex (right and left separately). REM-locked activation is widespread, but REM-locked activation peaks (which were identified after raising the statistical threshold to corrected $P < 0.00005$) are clearly localized in the structures listed above (Hong et al., 2009). Studying these structures with fUS that has high sensitivity and 10-100 ms temporal resolution (Deffieux et al., 2021)—enabling event-related assessment of the top-down directional propagation of signals in hierarchical processing after only a single trial of visual tasks (Dizeux et al., 2019)—will no doubt expand our knowledge of dreaming and waking consciousness. fUS studies employing rapid eye movement events as a probe will advance exploration of brain network dynamics unperturbed by external stimuli (i.e., while much of the external sensory input to the brain is blocked during REM sleep). The strategy of identifying notable brain structures to study with fMRI (which offers simultaneous measurement of the whole brain rather than plane-by-plane measurements) before single unit recording worked well to elucidate face recognition (Chang and Tsao, 2017).

We thank the reviewer for these positive comments. We fully agree that using the rapid eye movement tracking signal as a probe or beacon for fUS imaging would enable researchers to track directional connectivity based on lag correlations at high temporal resolution. It should become a powerful tool for the study of dreaming and waking consciousness. In future studies, before volumetric imaging becomes available, we will focus on systematically including the regions listed above, namely primary visual cortex, thalamic reticular nucleus, claustrum, cholinergic basal nucleus and retrosplenial cortex.

It was reported last month that the relatively small areas exempt from widespread REM-locked brain activation were restricted to the default mode network (Hong, Fallon, Friston, 2021). I would like to know if the authors observed in the default mode network a similar pattern of attenuation of the brain-wide 'vascular surge'. In particular, I would like to know if they observed attenuation of hyperemia in the left retrosplenial cortex during phasic REM sleep, as we observed in our fMRI study. Ferrier et al. (2020) reported sensory task-induced bilateral CBV decrease in granular retrosplenial cortex (RSG) (greater decrease in Lt > Rt). It is interesting to note that a solitary small area corresponding to the left RSG was identified as the 5th component by ICA.

We thank you for this very interesting and recent article we did not know of. We agree that we found a CBV decrease in RSC during sensory stimuli in mice in Ferrier et al. Moreover this decrease was higher in left RSC than right RSC. However, in the present experiments during REM sleep we did not see such a decrease. We carefully checked our experiments. The retrosplenial cortex is one of the most activated regions during the vascular surges. We did

not see any significant differences between the left and right RSC. You will see below some examples of CBV changes during REM sleep highlighting this bilateral increase in RSC during vascular surges for several animals. These have been included in the Supplementary Material.

Both fUS and fMRI measure hemodynamic changes. This fUS study of REM sleep observed vasodilation in the whole brain. This may illuminate our earlier observation of REM-locked vasodilation, indicated by a robust periventricular fMRI BOLD signal decrease time-locked to rapid eye movements (Hong, Fallon, Friston, 2021). It is unlikely that REM-locked vasodilation occurs only in the periventricular areas in humans. Rather, it may be that fMRI (due to its nature) can detect vasodilation only at the water-brain tissue border (because of partial volume effect) even though vasodilation occurs brain-wide in humans as well as in rats.

We humbly don't know how to explain these discrepancies between fMRI in humans and fUS imaging in rats. Nevertheless, as the amplitude of the BOLD fMRI signal is determined by the difference between the amount of blood flow increase, which increases the signal, and the use of O₂ by neurons, which reduces the signal, a potential explanation for the discrepancies between fUS imaging fMRI imaging results in RSC could be an overconsumption of O₂ during the vascular surges. For example, increased neuronal activity during seizures is associated with positive BOLD and increases in CBF in the cortex, but negative BOLD and decreased CBF in subcortical structures such as the striatum, despite equivalent LFP sizes (Mishra AM et al. 2011 Where fMRI and electrophysiology agree to disagree: corticothalamic and striatal activity patterns in the WAG/Rij rat. *J. Neurosci.* 31, 15 053– 15 06).

We agree with the reviewer in assuming that REM-locked vasodilation are unlikely to occur only at the water-brain tissue border and that such observation might be due to the intrinsic nature of BOLD fMRI. Additionally, we would like to insist on the fact that fUS and fMRI, though linked to hemodynamic changes, do not measure the same parameters. fUS imaging measures cerebral blood volume while BOLD measures a ratio between oxygenated and deoxygenated blood. These measures are not unequivocally linked such as the one cannot easily be inferred from the latter and vice versa. In particular, sensory-evoked CBV activations are faster than BOLD (Drew 2019). In particular, isolated optogenetic stimulation of pyramidal neurons causes a small increase in flow and a large consumption of oxygen, while isolated optogenetic stimulation of interneurons causes a small oxygen composition increase and large increases in flow (Vazquez et al. 2018). Though, again, we agree that REM-locked hemodynamic activations are probably distributed in large portions of the brain, the differences in the two techniques can account for these discrepancies. Future studies of REM activations with fUS will help resolve this important question.

Lines 102 – 105 “yet this hyperemic activity seems to be extremely energy consuming. We assume that if such energy-demanding activity has been maintained across evolution, it must have some important role for the survival of the animal, which does not seem to have been found yet.”

Lines 258 – 259 “Such hyperemic activity might be physiologically important as it was kept throughout evolution, despite its energy consumption.”

Consciousness is constructive and inferential. Generation of consciousness is linked to widespread brain activation, which is energy demanding, although the brain strives for energy efficiency (Hong, Fallon, Friston, Harris, 2018). The crux of consciousness generation—the world-model, with the self-model at its center—can be viewed as “a wonderfully efficient

control device” (Metzinger, 2009) enabling interaction with the world, which is essential for the survival of individuals and of the species. In fact this global hyperemic activity is linked to consciousness generation, which is essential for survival, this would justify its extremely high energy consumption.

We thank the reviewer for this great perspective and we added this reference and the associated comment in the discussion. The question of consciousness, especially in animals, is extremely hard to tackle. However, we would like to point out that in a behavioral task of repeated locomotion events where the animal is conscious, no activation with comparable amplitude and spatial extent has been observed in rats (Bergel et al. 2020). Thus, brain-wide activation probably is not directly associated with conscious experience.

Lines 67 and 69

The authors may consider adding the following paper in the reference: Hobson, J. A. (2009). REM sleep and dreaming: towards a theory of protoconsciousness. *Nat. Rev. Neurosci.* 10, 803–813. doi: 10.1038/nrn2716 “or the brain maturation specifically during early life REMS (Boyce et al., 2016; Hobson, 2009) and more precisely to aid sensorimotor system development (Hobson, 2009) through muscle twitches (Blumberg et al., 2013).” Related to this neurodevelopmental function of REM sleep, Hobson proposed that REM sleep provides off-line practice for walking in utero, before we actually learn to walk after birth (Hobson 2017, cited in Hong et al., 2018). [Hobson, A. (2017). *Conscious States: The AIM Model of Waking, Sleeping, and Dreaming*. Scotts Valley, CA: Create Space.]

We added this reference in the Introduction section Hobson, J. A. (2009).

Rats were used for the study, but ‘rat’ was not mentioned in the abstract.

This omission was corrected.

CBV was not defined. I assume that CBV stands for cerebral blood volume.

Yes, sorry for this, we added the definition of CBV

Figure 1 legend. “color: regional CBV traces” What color?

This was specified.

Figure 3D Gangliacampus -> Ganglia campus

This was corrected.

Table 1 -> Supplementary Table 1

This was corrected.

Reviewer #2 (Remarks to the Author):

The current work uses functional ultrasound imaging to examine and characterize brain-wide vascular responses to different stages of sleep in rats. The work extends previous work by researchers who are well-versed in both functional ultrasound and sleep research. While there are a few key take away points from the work, such as the dissociation of amygdalar activity from the rest of the brain during REMS, much of this work also overlaps with one of the authors' previous papers. While a valiant effort was put into this work, the analysis is fairly simple and shallow, and the authors do not present many significant and/or well-supported results. In its current state, I do not think this manuscript can be accepted for publication and needs significant work in terms of both the analysis and narrative. For this reason I am recommending rejection of this manuscript. I am sorry that I cannot be more supportive at this time, but wish the authors the best of luck in future submissions.

We thank the reviewer for his general comment. From your reading, we understand that we did not sufficiently emphasize the differences between this work and the previous publication from our group (Bergel et al. 2018, Nature Communications), both in terms of analysis and narrative. In light of this comment, we strongly modified Figures 1, 2 and 3, added new analyses and thoroughly reorganized the manuscript, in particular the introduction and discussion and addressed your general comments. We hope that these modifications will strengthen the quality of the current paper and highlight the differences with the previous paper.

In addition, we would like to insist on the fact that the main findings of this paper would not have been possible in the previous setup (single plane). The incremental step of scanning the brain over multiple coronal and parasagittal planes, together with the new analyses presented in the revised version of the manuscript, brings significant novel results (3D characterization of REMS networks, comparison between multiple brain regions between tonic and phasic REM sleep, large-scale characterization of vascular supply, anterior/posterior dissociation of amygdalar activity) that could not have been observed in the previous setup.

Please see my detailed points below:

1) I find the overall narrative of the current work in terms of the role of REMS in emotion regulation a bit grandiose. None of the experiments described in this work indicate any involvement of emotions, neither via observation nor via stimulus delivery. I feel that the need to elucidate REMS is a strong enough motivation for the current work, which also builds nicely off a previous study performed by many of the same authors. I wonder if the authors can simply expand this line of thought without incorporating emotions.

We agree that the role of emotions in REMS is not central to the current manuscript and that the need to elucidate REM sleep is a strong self-sufficient motivation for the current work. We thus have thoroughly rewritten the introduction. The role of REMS in emotion was toned down and only mentioned in the discussion.

2) The information provided in Figures 1 and 2 largely overlaps with that in one of the authors' previous works (Bergel et al. 2018 Nature Communications). From what I can tell, it seems that the main difference between Figures 1 and 2 in these two manuscripts is that the current

ones include information from a larger field-of-view. In this sense, I mean that Bergel et al. 2018 only examined a single plane and the current work examined multiple planes (coronal and sagittal), and therefore include more brain regions. While this additional information in theory represents a more comprehensive evaluation of these vigilance states, the analysis performed does not seem to teach the reader anything new that could not have been learned from the previous paper. The authors should explain the results from these figures in a clearer way and specifically describe how this information deviates from that in the previous work.

We agree that we did not sufficiently insist on the important differences between our previous work (Bergel et al 2018, Nature Communications) and the present study. Our former work was the first study using fUS imaging in sleeping rodents and was performed over a single coronal plane during REM sleep in 26 brain regions. It revealed a strong phasic hyperemia during REM sleep and the existence of tonic vascular surges during REM sleep, which correlated with hippocampal high gamma activity. However, this initial work posed numerous new questions. Was this vascular hyperemia present in the rest of the brain? Were vascular surges comparable across brain regions? What is the origin of blood supply to these events and how large arteries irrigate different regions of the brain? Such questions were the initial motivation to perform sequential scanning of the brain during REMS, not to merely increase the field of view. Additionally, as you mentioned, we insisted on performing parasagittal recordings to be able to simultaneously assess frontal and dorsal brain regions.

To explain our results in a clearer way and specify how the current project brings novel information that was not present in the previous paper, we replaced Figure 1 & 2 into two new figures incorporating novel analyses.

- In figure 1, we performed volume reconstruction of REMS networks and showed the strong activation of retrosplenial, anterior cingulate, dorsal thalamus, superior colliculus and dorsal hippocampus. We also compared quantitatively the level of activations in 56 brain regions, spanning $\frac{2}{3}$ of the total brain volume.
- In figure 2, we computed mean activation maps for all recordings (Figure 2C, Supplementary Figure 5) yielding a measure for the spatial recruitment (as a proportion of 'active pixels', see Methods) of regions during both tonic and phasic REMS (Figure 2D). We also quantified the heterogeneities of phasic events across the dorsoventral and anteroposterior axes of the brain (Figure 2E). This confirms that vascular events are stronger and longer in the posterior and medial part of the brain.

Last, we would like to insist on the fact that the current work describes the spatiotemporal dynamics of the vascular network in **more than 259 major brain regions representing 2/3 of the total rodent brain volume**, including most of the forebrain, midbrain and cortical structures **over hundreds of REMS episodes** (26 brain regions in 32 REMS episodes in the previous study). To date, this is the first resource to provide a direct precise quantification of spatiotemporal hemodynamics at the whole-brain scale at this level of detail during REM sleep and should be of considerable interest for the neuroscience community throughout the world. We also fully characterized blood supply during REM sleep and the spatiotemporal signatures of blood flow in the main cerebral arteries (Figure 4), which provides unique information about the vascular network during REM sleep and could become of particular interest for studies on brain drain during sleep due to the recent demonstration that brain drain in the glymphatic system is mainly driven by blood flow pulsatility in vessels.

3) I find the comparison between the tonic and phasic activation problematic for a few reasons. In general, the argument that various cortical structures are more correlated with phasic rather than tonic REMS is weakly supported by the results presented. To more clearly support this claim, I would expect at least a difference map between the two correlation maps presented in Figure 3C. Currently the correlation maps for tonic and phasic REMS look very similar and one has to look quite closely to see the differences pointed out by the authors. A difference map would clearly show instances of higher/lower correlation between the two conditions. In addition to this, I feel that a simple correlation analysis is not the strongest way to show the difference desired by the authors. A general linear model (GLM) analysis would more comprehensively and with greater statistical power evaluate potential differences in tonic vs phasic activation. This brings me to an additional point of how the tonic and phasic regressors (or seeds) were constructed. I don't understand why the phasic regressor would also contain tonic information, and would rather think that two binary regressors of only zeros and ones (1 phasic REMS, 0 otherwise and 1 tonic REMS, 0 otherwise) would be more appropriate. This may also be necessary for a GLM analysis as the current regressor designs (as depicted in Figure 3A) seem highly collinear and could lead to unstable GLM results. Nevertheless, with orthogonal regressors (as I describe above), the authors should be able to obtain stable tonic and phasic REMS maps and further be able to obtain a contrast map of tonic vs phasic REMS. I strongly encourage the authors to explore this option.

We thank the reviewer for this point and agree that the analysis does not clearly show the stronger involvement of cortical structures into phasic REM sleep, in comparison with other regions. We also agree that the previous choice of regressors (not orthogonal) is somewhat unnatural. We have thus computed orthogonal binary variables as proposed (non-zero during REM-phasic and REM-tonic periods respectively, see Figure 2B) and computed corresponding correlation maps, as well as the difference maps in Figure 3A-B. These clearly show the dissociation between cortical structures and subcortical brain structures.

In addition, we also complemented the analysis and performed a General Linear Model analysis and computed the difference maps. The corresponding results are presented in Figure 3C and clearly show that regressors associated to tonic activity are more important in subcortical structures whereas phasic regressors are more prominent in midbrain and cortical structures.

4) The amygdalar phenomenon is quite interesting and seems convincing in terms of the connectivity analyses in Figure 5 A, C and D. However, the time series information presented in Figure 5B doesn't portray as strong of an argument. It does appear that the amygdalar sub-regions do deviate from the whole brain trace, but only slightly. To determine how meaningful this deviation is, it would be useful to also include a time series trace from a region(s) that highly correlates with the whole brain signal. In this sense, the reader could clearly see by eye what relatively high and low correlated activity looks like.

We thank the reviewer for this comment and we added this information in the revised manuscript.

Furthermore, the authors present the results of very small amygdalar sub-regions, but do not discuss how reliable the segmentation of these areas are. While it is generally understood the functional ultrasound exhibits a high spatial resolution, I cannot find the pitch of the probe

stated anywhere in the methods. This lack of specified resolution, combined with no scale bars shown in the images in Figure 5C-E make it difficult to assess whether it is reasonable that functional ultrasound can resolve structures at the spatial scale being claimed. I would further assume that errors from the manual registration process and variability in window width/quality could further complicate the accurate identification of these regions. It would be useful if the authors were to clarify these important pieces of information.

We are sorry for the omission of the pitch information. The spatial pitch of the probe is 100 μm and the in plane pixel resolution is 100 μm x 100 μm . We corrected this omission in the revised manuscript. Each subregion of the amygdala contains in average 25 to 35 pixels (see for example Fig 5E). Data presented in figure 5 are raw data and did not require any spatial filtering or smoothing. We agree that the registration part is not an easy task and could be subject to additional variance in the estimations. Here, the registration was performed by highly trained neuroscientists using dedicated vascular landmarks and recent works by our group have shown for similar configurations in mice (Nouhoum et al., 2022) that the typical registration precision is of the order of 1 to 2 pixels size (typically less than 200 μm for manual registration and it could even be less with an automated registration based on the recognition of the vascular print of individual animals). We added this reference and comments in the discussion in order to address this point. We are not able to certify that the registration is precise at the voxel size. However, for these brain structures of median size, we are very confident that the registration is correctly performed thanks to the 3D vascular landmarks approach.

REVIEWERS' COMMENTS:

Reviewer #1 (Remarks to the Author):

It is clear that the authors invested a significant amount of time and effort in producing a major revision. The manuscript has improved substantially. I have minor suggestions only.

Page 6

“(see Table 1 and Supplementary Figure S1 and S2 for data acquisition, and Supplementary Figure S3 for electrode implantation and identification).”

I could not find Table 1. Shouldn't it be supplementary Table 1?

Page 10

“Silent (Figure 5B, second part of the episode) .”

The space before the period should be removed.

Page 15

“In addition, fMRI studies have shown that the presence of rapid-eye-movement was strongly correlated with the activation of many brain regions such as the oculomotor circuit, the corticothalamic sensory system, the language system or even the cholinergic and also serotonergic (Hong et al., 2009).”

I suggest that “and also serotonergic” be removed. The periventricular fMRI BOLD signal decreases (PVSD) time-locked to rapid eye movements in sleep (REM) was initially explained by REM-locked serotonergic deactivation (Hong et al. 2009). Later, REM-locked PVSD was explained by REM-locked periventricular vasodilation, i.e., non-serotonergic mechanism (Hong et al. 2021). I also suggest that system be added after “even the cholinergic”.

Page 11, First paragraph in Discussion.

The authors noted that 3D fUS imaging requires head restraint but 2D fUS imaging does not.

It appears that head restraints which are required for fMRI suppress REM sleep in humans (Hong et al., 2009).

Page 15

Two references were added: REM-locked widespread activation is essential for generation of dream consciousness (Hong et al. 2018); and secondly, global brain activation is essential for generation of waking consciousness (Dehaene and Changeux, 2011).

Adding these interpretations of the brain-wide hyperemia in phasic REM sleep strengthened the manuscript.

Page 15

Dissociation of the amygdala against the backdrop of brain-wide activation during REM sleep is a novel and very interesting finding. Adding “another hypothesis” to explain this novel, interesting finding strengthens the manuscript. This added hypothesis is further supported by the following influential

theories: 'theory of constructed emotion' (Barrett 2017) and 'higher order theory of emotional consciousness' (LeDoux and Brown 2017) — for fear to be consciously experienced, information from the amygdala should be available to the brain-wide processing that enables conscious experience.

Page 16

Importantly, the authors mentioned a new camera set-up for timing rapid eye movements in their future studies to demonstrate the link between vascular surges and rapid eye movements. Future studies will be improved by this set-up. As rapid eye movements (REMs) are temporally precise events, timing of REMs allows time series analysis which is statistically efficient. Hong et al. (2018) noted, "Our findings suggest that REM-locked brain activity is distinct from baseline brain activity during phasic REM sleep. Clearly, REM-locked brain activity changes overlap with those associated with REM sleep (particularly, phasic REM sleep, which entails most REMs); however, our findings indicate that the overlap may reflect spillover effects accompanying REMs. In particular, brain activity changes time-locked to video-timed REMs speak to the utility of characterizing temporally precise REMs, as opposed to comparing the neural correlates of phasic vs. tonic REM sleep."

Including the timing of rapid eye movements in future fUS studies will advance the research goal of the authors – exhaustive characterization of REMS-associated global hyperemia during rodent REMS and its deeper understanding. Moreover, timing rapid eye movements will enable a more straightforward comparison of animal data with human data. Their powerful fUS technology, using REMS as a probe, has the potential to make significant contributions to our understanding of the brain. I am very much looking forward to seeing the results of their future studies.

Reviewer #2 (Remarks to the Author):

I want to start by commending the authors on their efforts in not only re-analyzing a significant portion of their data but also in re-writing much of the manuscript. My previous comments have been mostly addressed and I think the manuscript has been markedly improved with a more reasonable narrative and claims that are now better supported by clearer results. I have only a few minor comments that I think could further explain a few key points of the manuscript and make the results even clearer. Please see my detailed comments below.

1. The threshold for defining active and inactive pixels is a bit confusing and unclear (page 7). How exactly is this threshold defined to classify pixels classified as active or inactive? Were the fUS data normalized in some way (perhaps to a pre-REM episode baseline) that resulted in the "pixel activity" units, which were then thresholded at XX%? This definition sets the stage for the entire phasic vs tonic REM temporal segmentation and should be explained a bit better.

2. I noticed this observation in the original submission, but it did not seem as apparent then. The atlas brain slices that the authors show are often off-center and/or cut off. It would be convenient to the reader if all of the slices were aligned so that each one is visualized in its entirety. Based on Figure 4A and the inherent variability in the cranial window procedure, it is understandable that there will be a different amount of recording coverage for different regions. "Gray-ing" out these low-coverage regions/pixels (defined by a threshold of how many sessions/animals they were observed in), but still showing the full outline of the slice would be ideal so that the reader gets the full picture. In many cases – Figure 3a in particular - the abrupt change in color near the lateral sides of the brain does not do the data justice and indicates information that may simply be a result of low data coverage. I think it would be in the authors' best interest to clarify this and show information that is only robust across the majority of their recordings.

3. Please add units to the color bar in Figure 3c.

4. It would be nice to highlight in some way the rows/columns for the anterior and posterior amygdala in figure 5a.

Answer to the reviewers:

REVIEWERS' COMMENTS:

Reviewer #1 (Remarks to the Author):

It is clear that the authors invested a significant amount of time and effort in producing a major revision. The manuscript has improved substantially. I have minor suggestions only.

Page 6

“(see Table 1 and Supplementary Figure S1 and S2 for data acquisition, and Supplementary Figure S3 for electrode implantation and identification).”

I could not find Table 1. Shouldn't it be supplementary Table 1?

Indeed, “supplementary” has been added.

Page 10

“Silent (Figure 5B, second part of the episode).”

The space before the period should be removed.

Space removed.

Page 15

“In addition, fMRI studies have shown that the presence of rapid-eye-movement was strongly correlated with the activation of many brain regions such as the oculomotor circuit, the corticothalamic sensory system, the language system or even the cholinergic and also serotonergic (Hong et al., 2009).”

I suggest that “and also serotonergic” be removed. The periventricular fMRI BOLD signal decreases (PVSD) time-locked to rapid eye movements in sleep (REM) was initially explained by REM-locked serotonergic deactivation (Hong et al. 2009). Later, REM-locked PVSD was explained by REM-locked periventricular vasodilation, i.e., non-serotonergic mechanism (Hong et al. 2021). I also suggest that system be added after “even the cholinergic”.

Corrections made.

Page 11, First paragraph in Discussion.

The authors noted that 3D fUS imaging requires head restraint but 2D fUS imaging does not.

It appears that head restraints which are required for fMRI suppress REM sleep in humans (Hong et al., 2009).

Information added to the discussion.

Page 15

Two references were added: REM-locked widespread activation is essential for generation of dream consciousness (Hong et al. 2018); and secondly, global brain activation is essential for generation of waking consciousness (Dehaene and Changeux, 2011).

Adding these interpretations of the brain-wide hyperemia in phasic REM sleep strengthened the manuscript.

Page 15

Dissociation of the amygdala against the backdrop of brain-wide activation during REM sleep is a novel and very interesting finding. Adding “another hypothesis” to explain this novel, interesting finding strengthens the manuscript. This added hypothesis is further supported by the following influential theories: ‘theory of constructed emotion’ (Barrett 2017) and ‘higher order theory of emotional consciousness’ (LeDoux and Brown 2017) — for fear to be consciously experienced, information from the amygdala should be available to the brain-wide processing that enables conscious experience.

These references have been added to the discussion.

Page 16

Importantly, the authors mentioned a new camera set-up for timing rapid eye movements in their future studies to demonstrate the link between vascular surges and rapid eye movements. Future studies will be improved by this set-up. As rapid eye movements (REMs) are temporally precise events, timing of REMs allows time series analysis which is statistically efficient. Hong et al. (2018) noted, “Our findings suggest that REM-locked brain activity is distinct from baseline brain activity during phasic REM sleep. Clearly, REM-locked brain activity changes overlap with those associated with REM sleep (particularly, phasic REM sleep, which entails most REMs); however, our findings indicate that the overlap may reflect spillover effects accompanying REMs. In particular, brain activity changes time-locked to video-timed REMs speak to the utility of characterizing temporally precise REMs, as opposed to comparing the neural correlates of phasic vs. tonic REM sleep.”

Including the timing of rapid eye movements in future fUS studies will advance the research goal of the authors – exhaustive characterization of REMS-associated global hyperemia during rodent REMS and its deeper understanding. Moreover, timing rapid eye movements will enable a more straightforward comparison of animal data with human data. Their powerful fUS technology, using REMS as a probe, has the potential to make significant contributions to our understanding of the brain. I am very much looking forward to seeing the results of their future studies.

Reviewer #2 (Remarks to the Author):

I want to start by commending the authors on their efforts in not only re-analyzing a significant portion of their data but also in re-writing much of the manuscript. My previous comments have been mostly addressed and I think the manuscript has been markedly improved with a more reasonable narrative and claims that are now better supported by clearer results. I have only a few minor comments that I think could further explain a few key points of the manuscript and make the results even clearer. Please see my detailed comments below.

1. The threshold for defining active and inactive pixels is a bit confusing and unclear (page 7). How exactly is this threshold defined to classify pixels classified as active or inactive? Were the fUS data normalized in some way (perhaps to a pre-REM episode baseline) that resulted in the “pixel activity” units, which were then thresholded at XX%? This definition sets the stage for the entire phasic vs tonic REM temporal segmentation and should be explained a bit better.

Initial text (page 7) “By thresholding vascular activity in all brain pixels during REMS (which was set independently for each pixel based on the mean and standard-deviation of its distribution during active wake, see Methods), we were able to classify pixels as ‘active’ or ‘inactive’ during a single REMS episode, which allowed us to define time intervals as REM-PHASIC periods, that we also refer to as vascular surges (VS), when more than 50% of brain pixels were active simultaneously for more than 3 seconds (Figure 2B)”

Update Methods: CBV maps & spatial averaging & activation threshold

Each voxel was normalized independently before performing spatial averaging. The activation threshold used to classify pixels as active or inactive during REMS episodes, was set as $\mu_{AW} + n \cdot \sigma_{AW}$, where μ_{AW} and σ_{AW} are respectively the mean and standard deviation of pixel CBV signal after normalization (expressed as %CBV change). n was set to 1. This means that a pixel was considered ‘active’ when the difference between its value and the mean of the AW distribution was greater than one standard-deviation of the AW distribution. This definition imposed that pixel activity outmatch active wake levels to be considered ‘active’ which is significantly more stringent than using the mere QW distribution.

2. I noticed this observation in the original submission, but it did not seem as apparent then. The atlas brain slices that the authors show are often off-center and/or cut off. It would be convenient to the reader if all of the slices were aligned so that each one is visualized in its entirety. Based on Figure 4A and the inherent variability in the cranial window procedure, it is understandable that there will be a different amount of recording coverage for different regions. “Gray-ing” out these low-coverage regions/pixels (defined by a threshold of how many sessions/animals they were observed in), but still showing the full outline of the slice would be ideal so that the reader gets the full picture. In many cases – Figure 3a in particular - the abrupt change in color near the lateral sides of the brain does not do the data justice and indicates information that may simply be a result of low data coverage. I think it would be in the authors’ best interest to clarify this and show information that is only robust across the majority of their recordings.

The suggested modifications have been added to Figure 2C, 3A and 3C.

3. Please add units to the color bar in Figure 3c.

The color bar units have been added to Figure 3C.

4. It would be nice to highlight in some way the rows/columns for the anterior and posterior amygdala in figure 5a.

A square has been added to highlight the anterior and posterior amygdala.